# Gesture encoding in human left precentral gyrus neuronal ensembles
Carlos E. Vargas-Irwin [1,2,3] ✉, Tommy Hosman [3,4], Jacob T. Gusman [2,3,4,5], Tsam Kiu Pun [2,4,5], John D. Simeral [2,3,4], Tyler Singer-Clark [6], Anastasia Kapitonava [6], Claire Nicolas [6], Nishal P. Shah [7], Donald T. Avansino [8], Foram Kamdar [7], Ziv M. Williams [9], Jaimie M. Henderson [7,10,11] & Leigh R. Hochberg [2,3,4,6]

Understanding the cortical activity patterns driving dexterous upper limb motion has the potential to benefit a broad clinical population living with limited mobility through the development of novel brain-computer interface (BCI) technology. The present study examines the activity of ensembles of motor cortical neurons recorded using microelectrode arrays in the dominant hemisphere of two BrainGate clinical trial participants with cervical spinal cord injury as they attempted to perform a set of 48 different hand gestures. Although each participant displayed a unique organization of their respective neural latent spaces, it was possible to achieve classification accuracies of ~70% for all 48 gestures (and ~90% for sets of 10). Our results show that single-unit ensemble activity recorded in a single hemisphere of human precentral gyrus has the potential to generate a wide range of gesture-related signals across both hands, providing an intuitive and diverse set of potential command signals for intracortical BCI use.

The human hand is composed of 27 bones linked to 15 extrinsic and 11 intrinsic muscles acting across 18 joints, affording an enormous number of possible movements across more than 20 degrees of freedom[1,2]. Prominent connections to the hand's spinal motor neurons from sophisticated neural control systems in the cerebral cortex imbues the hand with remarkable dexterity[3]. Indeed, manual dexterity in mammals is well correlated with the expansion of the neocortex[4]. Hand movements engage a disproportionately large constellation of cortical regions in primates[5–7], with the precentral gyrus (PCG) providing one of the most direct and prominent sources of corticospinal projections in humans[8]. Diseases or disorders that disrupt neural command signals driving hand motion can have devastating results, limiting the ability to perform everyday tasks. For people with tetraplegia, regaining arm/hand function is a top priority towards improvement in quality of life[9–12]. In this study, we aim to examine how information related to discrete hand gestures is encoded by neuronal ensembles in the human PCG and evaluate the potential use of these signals for intracortical brain-computer interface (iBCI) applications.

Emerging iBCI technology offers the possibility of building new links between the nervous system and the external world, bypassing damaged motor pathways[13]. The goals for intracortical BCIs include restoring reaching and grasping actions[14–17] as well as control over communication devices such as tablet computers using continuous 2D 'point and click' control[18–20], decoding attempted handwriting movements and converting them to text in real time[21], and even controlling a quadcopter with continuously decoded digit movements over 4 dimensions[22]. Discrete decoding of individuated finger movements of one hand has been demonstrated in humans using a variety of signals, including EEG[23,24], fMRI[25], and ECoG[23,26–30]. Single-unit activity in non-human primates has also been used to discriminate between 12 individuated finger/wrist movements[31,32] as well as the continuous independent movement of two finger groups[33–35]. Analysis of cortical single-unit activity in human participants has been limited to relatively small sets of between five and seven hand gestures[36–39]. Although all these results highlight the high informational content of cortical areas related to dexterous hand movements, so far the neural representation of only a small subset of these actions has been explored. This restricted scope

[1]Department of Neuroscience, Brown University, Providence, RI, USA. [2]Robert J. and Nancy D. Carney Institute for Brain Science, Brown University, Providence, RI, USA. [3]VA Center for Neurorestoration and Neurotechnology, VA Providence Healthcare System, Providence, RI, USA. [4]School of Engineering, Brown University, Providence, RI, USA. [5]Biomedical Engineering Graduate Program, School of Engineering, Brown University, Providence, RI, USA. [6]Center for Neurotechnology and Neurorecovery, Department of Neurology, Massachusetts General Hospital, Harvard Medical School, Boston, MA, USA. [7]Department of Neurosurgery, Stanford University, Stanford, CA, USA. [8]Howard Hughes Medical Institute at Stanford University, Stanford, CA, USA. [9]Department of Neurosurgery, Massachusetts General Hospital, Harvard Medical School, Boston, MA, USA. [10]Wu Tsai Neurosciences Institute, Stanford University, Stanford, CA, USA. [11]Bio-X Institute, Stanford University, Stanford, CA, USA. ✉e-mail: Carlos_Vargas_Irwin@Brown.edu

not only limits the practical applications of gesture decoding but also hinders our understanding of the broader neural mechanisms underlying dexterous hand movements. There is a need to explore the neural representation of the cortical area related to a wider range of dexterous hand gestures, which could potentially be harnessed for a more versatile iBCI system.

The present study examines the neural activity of ensembles of human motor cortical neurons across a set of 48 different gestures engaging the digits, wrists, and forearms for the right and left limbs in two participants. In addition to providing a benchmark for gesture classification accuracy using human intracortical signals, this study also introduces a novel way to organize gestures into functional groups based on the intrinsic similarity of their associated neural activity patterns. This bottom-up approach aims to map the intrinsic structure of the neural latent space for gestures, and examine whether latent space geometry is conserved across study participants. Our results reveal subject-specific organization that nevertheless displays similar informational content widely distributed among individual neural activity features with heterogeneous encoding properties reflecting a large number of potential movement synergies. These findings underscore the potential of intended gestures across both limbs as control signals for BCI applications, even when neural recordings are constrained to a single hemisphere.

## Results

We recorded neural data under two control conditions: open-loop (where the participant was asked to perform hand gestures following cues on the screen while neural data was recorded without providing feedback) and closed-loop (where the neural signals were used to decode intended gestures, and the results were presented to the participant on the screen). Each open-loop block included 2 repetitions of each of the 48 gesture types, randomly intermixed. At least three such blocks were included at the beginning of each session. Data collected during these initial open-loop blocks was used to train the decoders used for closed-loop control. Decoding accuracy tends to decrease as the number of gesture classes increases. To address this challenge, we structured closed-loop control blocks to include sets of six or seven gestures at a time. This allowed us to collect neural activity patterns representative of the full set of 48 gestures in a reasonable amount of time while providing sufficient decoding accuracy to keep the participants engaged. As described below, we performed additional offline analysis combining data across multiple sessions (including both control conditions) in order to build more comprehensive models of gesture-related activity for the full set of gestures examined.

Neural recordings were collected from two participants: participant T11 (six sessions) and participant T5 (two sessions). Each participant had two 96-channel intracortical microelectrode arrays placed chronically into the left PCG as part of the ongoing BrainGate2 pilot clinical trial (see "Methods"). Neural features used for closed-loop control and offline analysis included spiking activity (threshold crossings; RMS < −3.5) as well as local field potential power in the 250 Hz–5 kHz band (8th order IIR Butterworth) for each microelectrode (for a total of 192 features per array, and 384 features evaluated per participant).

### Neural features in human PCG are modulated by multiple attempted gestures

Participants were asked to attempt a set of 48 different gestures (Fig. 1A). They were provided with images of a hand and timing cues presented on a screen as part of an instructed delay paradigm (Fig. 1B, see "Methods" for additional task details). At the beginning of each trial, a rectangular target appeared at the top and fell toward the bottom of the screen, accompanied by an image of the cued gesture and descriptive text. The participant was instructed to attempt the gesture as long as the target intersected the horizontal line displayed near the bottom of the screen ("attempt line"). For some sessions (3 for T11, and both sessions for T5), an additional "do nothing" prompt was added as a control (for a total of 49 different sampled states). We evaluated task-related activity in the following way: First, we

compared the values for each neural feature between the inter-trial interval (baseline) and each attempted gesture ($p < 0.001$, Wilcoxon Rank-Sum, see "Methods" for details). In this way, we determined the number of gestures producing task-related activity for each neural feature. The gesture task robustly engaged cortical activity in the PCG, with over 65% of the 384 neural features evaluated for each participant displaying task-related activity for at least one of the gesture types examined, and over 20% for at least 20 different gestures (Fig. 1C). Next, we examined how the neural activity patterns differed across the set of attempted movements.

### Closed-loop decoding individual gestures from neural activity

Closed-loop decoding was performed using groups of six or seven gestures at a time to make the task easier and maintain participant engagement. Each set included three gestures randomly selected for each hand; for blocks of seven gesture classes the 7th class was always "Do nothing". For this report, we define successful trials for closed-loop control as those where the most frequently decoded gesture during the 1 s attempt period (evaluated in 50 time bins) matched the cued gesture. Trials where no gesture was decoded were counted as incorrect. Closed-loop decoding was performed using multiclass linear discriminant analysis (LDA) followed by a hidden Markov model (HMM)[40] (See "Methods" section for details). For participant T11, mean decoding accuracy was 81.7 ± 11.8% for blocks with six classes, and 76.79 ± 10.6% for blocks with seven classes (Fig. 2A, left). For Participant T5, mean decoding accuracy was 71.4 ± 10.5%, and 78.8 ± 11.7% for six and seven-way classification, respectively (Fig. 2A, right). Overall, T5 displayed relatively highly accurate decoding for the "no action" class, improving results for 7-class decoding (Fig S3). The rest of the analysis was performed offline combining data from all recording blocks in order to examine the relationship between the full set of 49 gesture states.

### dPCA reveals varying levels of gesture-related variance across participants

To estimate the informational content and dimensionality of the neural features we used demixed principal component analysis[41], (see "Methods" for additional details). dPCA is a semi-supervised dimensionality reduction technique that finds task related components that capture a large percent of the population variance. The top 20 dPCA components capture nearly as much variance as the top 20 PCA components for participants T5, and T11, respectively (Fig. 3A, E). Here we find components that capture the response variance that is common across conditions (Common, Fig. 3B, F), components that capture variance related to the attempted gesture (Gesture, Fig. 3C, G), and components that capture variance related to the hand used (Hand/Lateral, Fig. 3D, H). Note that changes in the common component typically reflect the overall timing of task events, including anticipation of upcoming cues. Changes in these components can therefore precede the appearance of instructional cues, but do not reflect either effector or gesture information (as evidenced by the overlap across categories observed in Fig. 3B, F). Gesture and hand information components, by contrast, separate by category after cues are provided, reflecting the delay periods employed for each participant (Fig. 3C, D, G, H). Despite similar closed-loop decoding performance, the distribution of variance captured by these three groups was distributed differently for participant T5 and participant T11 (Fig. 3A, E pie charts). While both participants have a similar proportion of variance common across all conditions (38.6% T5, 42.2% T11), their fraction of gesture and hand variance explained is markedly different. The Hand components for T5 explain approximately 50% more variance than the Gesture components (33.6 vs. 22%). By contrast, T11's Gesture components explain over an order of magnitude more variance than the Hand components (50.2% vs. 4.8%).

### Data-driven bottom-up mapping of the neural latent space of gestures

The next phase of our analysis focused on generating latent spaces using similarity-based metrics[42] to examine the relationship between neural activity patterns associated with attempted gestures (See "Methods" for

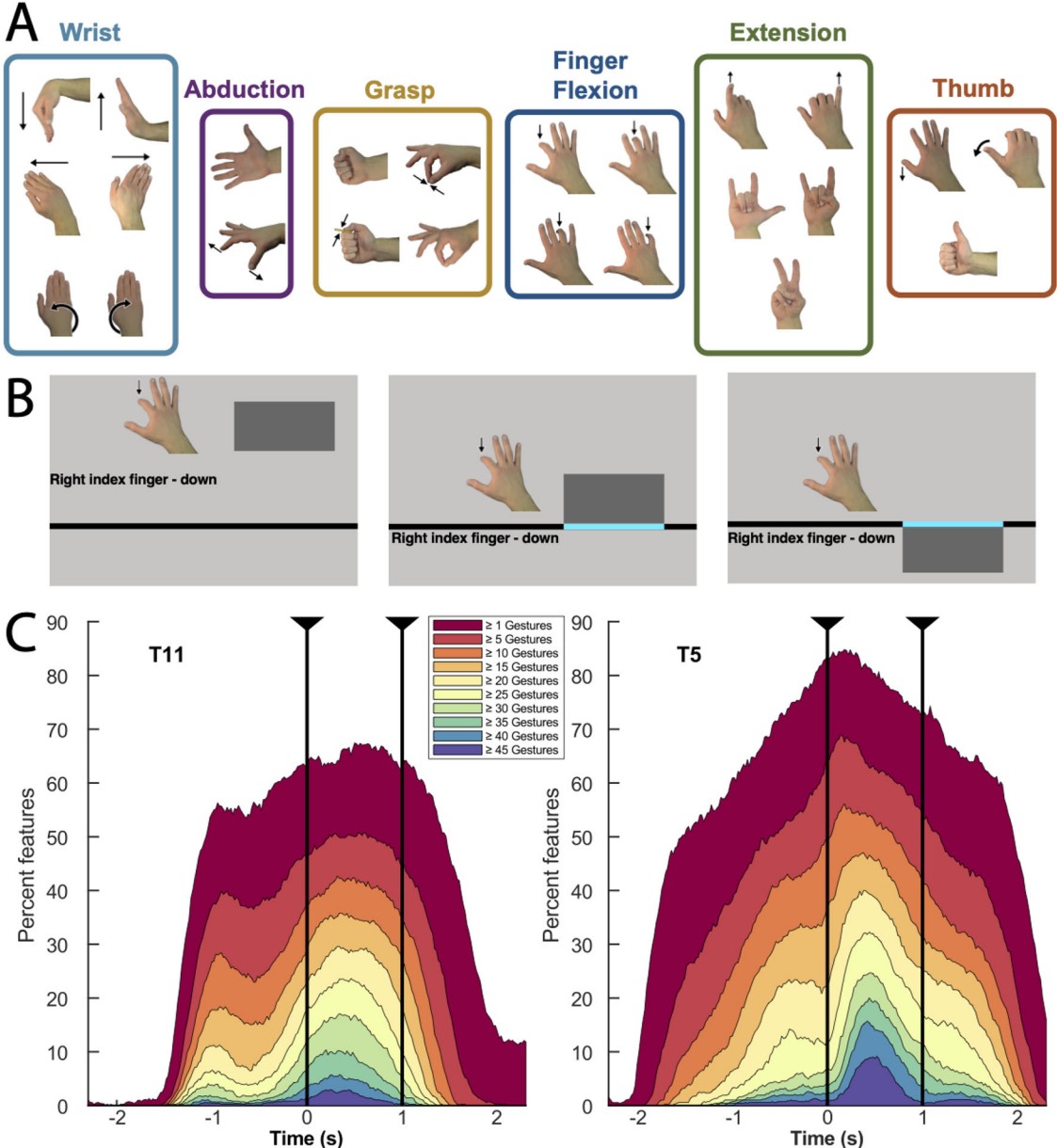

**Fig. 1 | Percentage of neural features displaying significant activity during the gesture task. A** Set of gestures presented (only right-hand gestures are shown, mirror images were shown for the left). Highlighted groups are derived from neural data, as described in the following sections. **B** Events in the cued gesture task timeline: 1. Cue onset (beginning of the instructed delay), 2. "Go" cue (initial crossing of the horizontal line by the falling box), and 3. Relax cue (departure of the falling box from the horizontal line). **C** Each curve shows the percentage of features displaying task-related activity (i.e., significantly different values from inter-trial interval) for different numbers of gesture conditions throughout the course of the

task ($p < 0.001$, Wilcoxon Rank-Sum, see "methods" for details). For example, for participant T5 ~9% of features diverged from baseline values for >45 gestures shortly after the "go" cue (dark blue curve). Around the same time, ~85% of features diverged from baseline for at least one gesture (dark red curve). Data were combined across sessions. The time axis is centered on the Go cue (time zero). Vertical lines mark the interval between the Go and Relax cues. Note that delay periods of different duration were used for each participant: for T11, cue onset (the appearance of the instructed gesture, prior to the "go" cue at time zero) was either 1 or 2 s before the go cue, for T5 the delay was 3 s.

details). Unlike dPCA, this approach does not require pre-defining marginalizations based on specific categories of interest, and is entirely based on the intrinsic similarity between the activity patterns. Our analysis focused on activity patterns associated with individual gestures taken from a 2 s window centered on the go cue (Fig. 1). Note that this time window includes activity related to movement planning as well as attempted movement execution. Activity related to movement planning has been widely recognized in motor and premotor cortex[43], and has been interpreted as adjusting the state of a dynamical system whose evolution ultimately results in cortical outputs driving movement[44]. Data was smoothed using a 200 ms Gaussian kernel. Similar results were obtained using different window durations, offsets, and

smoothing parameters (supplementary Fig. S1). We chose to use 15 dimensions for the latent spaces based on intrinsic dimensionality estimates obtained from dPCA, choosing values close to the asymptote (Fig. 3).

Latent spaces for participant T11 were generated independently for each recording session (with between 1232 and 868 gestures each) and then aligned using a shape-preserving landmark-based procedure[45], resulting in a single matrix representing 7322 individual gestures as 15D vectors. Subsequent analysis was performed using the entirety of the data for participant T11 aligned in this way. In order to examine the relationship between gesture-related activity patterns, we calculated the centroid for each of the 49 gesture classes in the combined latent space and generated a dendrogram

**Fig. 2 | Closed-loop decoding accuracy. A** Average decoding accuracy for sets of six (blue) or seven (orange) gestures. Each dot represents the accuracy for a gesture set (i.e., a single block of trials) in a given session. Dotted lines indicate expected chance levels. Error bars indicate standard deviations. Confusion matrices for two sample blocks (with rows representing true classes and columns representing decoded classes), representative of average decoding accuracy for one block in participant T11 (**B**) and T5 (**C**). Note that the confusion matrices count trials where no gesture exceeded the likelihood threshold as errors ("no decode", rightmost column). Additional confusion matrices for other gesture sets are included in Supplementary Figs. S2 and S3.

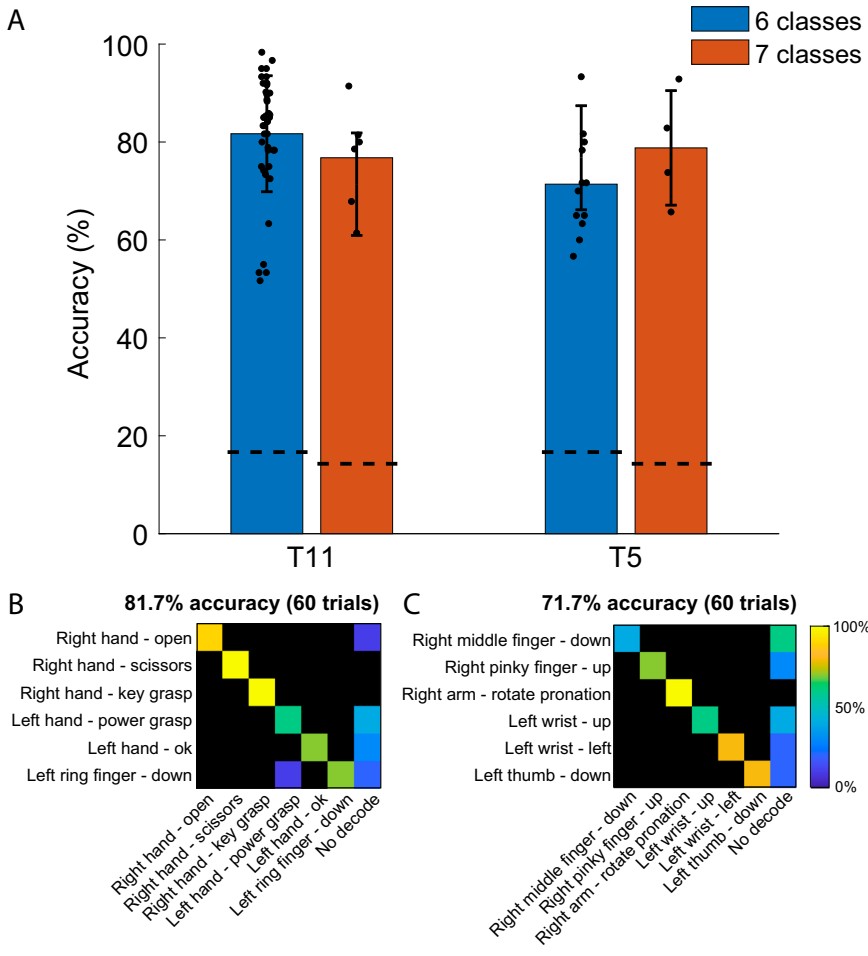

using Ward's method (Fig. 4A). Examining the dendrogram for participant T11 revealed six broad families of neural responses, associated with specific groups of gestures (Fig. 4A):

1. Wrist movements (flexion, extension, pronation, supination, and ulnar/radial deviation)
2. Thumb movements (with the thumb moving independently or in combination with finger flexion)
3. Individual Finger Flexion Movements
4. Grasping movements (including power grasp as well as movements involving opposition of the index and thumb)
5. Finger extension Movements (individual finger extension in isolation or combined with movement in other digits)
6. Digit Abduction (splaying the fingers or performing a "zoom-in" gesture)

Note that for participant T11, gestures performed with the dominant right hand (contralateral to the recording site) tended to be most similar to the same gesture performed with the left hand.

A similar procedure was used to generate a latent space using both sessions from participant T5 (with 944 and 962 gestures each, for a total of 1906 gestures). The latent space for T5 (Fig. 4B) differed in two key points. First, the most prominent feature was the separation between right and left hand gestures, in sharp contrast to T11. Second, the clear separation between the six gesture groups identified for T11 was not observed in the dendrogram for T5 (although some groups, e.g., finger flexion, were still partitioned into independent branches).

In order to further characterize the trends observed in the gesture dendrograms, we examined individual trials within the 15D gesture space (Fig. 5). We evaluated the separation between gesture groups using a nearest neighbor classifier (implemented with 10-fold cross-validation). For T11,

the six gesture groups and the "do nothing" control could be classified with over 94% accuracy, well above the expected chance value of 14.2%. Gesture group classification results for participant T5 also exceeded expected chance levels. As with T11, classification accuracy for finger flexion, wrist movements, and 'do nothing' exceeded 92%. However, there was a greater degree of overlap between the thumb, grasp, abduction, and extension groups, with accuracy ranging from 69 to 79% overall.

**Individual gesture classification**

Classification accuracy was evaluated over the full set of 48 individual gestures and the "do nothing" control condition. Note that this analysis was performed offline, combining data collected from all available open and closed-loop control blocks. As described in the previous section, classification was performed using a nearest neighbor classifier (implemented with 10-fold cross-validation). Remarkably, despite the different organization of the latent spaces, decoding accuracy was relatively similar across the two participants, with approximately 74% accuracy for T11 and 69% for T5 (Fig. 6).

We additionally examined gesture classification for each hand independently, with 24 gestures for each one (Fig. 7). For participant T11, there was a statistically significant ($t$-test; $p < 0.01$) improvement in classification accuracy for the right (dominant) hand (average 84% correct across all gestures) relative to the left (79%). However, results for both hands far exceeded expected chance levels (~4.2%). For participant T5, classification accuracy did not display significant differences across effectors, with ~69% average correct classification accuracy for the right and left hands.

The separability between right and left-hand actions (i.e., predicting the intended effector independent of the gesture attempted) was evaluated independently. Although for participant T11 right and left hand versions of

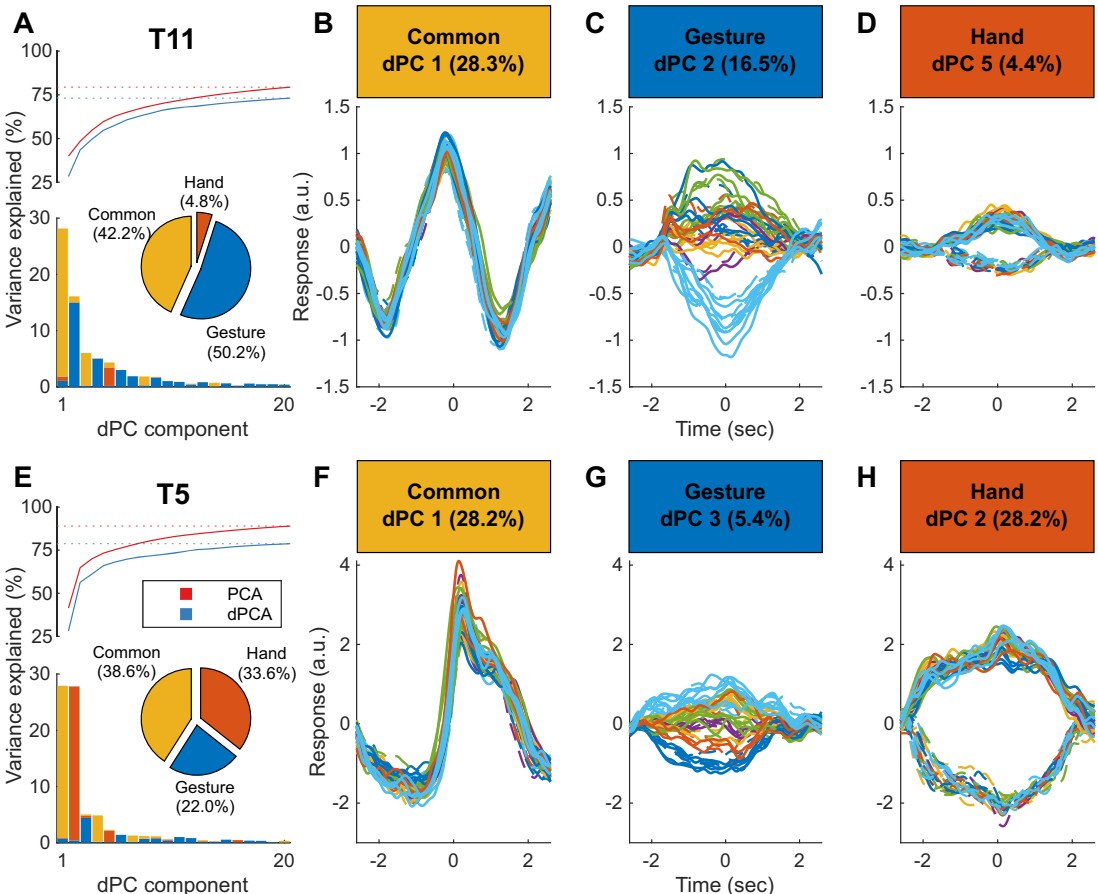

**Fig. 3 | Distribution of task-related information for neural ensembles.** The distribution of variance explained by decomposing ensemble activity into a set of components that correspond to three task-related categories, common across conditions (Common), cued gestures (Gesture), left vs right hand (Hand), using demixed principal component analysis (dPCA, see methods). **A–D** relate to participant T11 (across six sessions) and **E–H** relate to participant T5 (across two sessions). **A, E** The bars show the percent variance explained for the top 20 dPCs. The traces show the cumulative explained variance for the top 20 PCA and dPCA

components. The inset pie chart shows the distribution of variance explained per category across all 20 dPCs. **B, F** The response of ensembles projected into the top Common dPC. Each line is the projected trial average response for a single condition. Lines are colored by one of six gesture groups (identified in the following section, see Fig. 1), excluding the "do nothing" control condition. Dashed and solid lines reflect left and right-handed gesture conditions, respectively. **C, G** Same as (**B, F**) but ensembles are projected into the top Gesture dPC. **D, H** same as (**B, F**) but ensembles are projected into the top Hand dPC.

a given gesture tended to occupy adjacent sections of the latent gesture space (Fig. 5A), it was still possible to separate them well above chance levels, with ~89% accuracy. Participant T5 displayed a wider separation between effectors (Fig. 5B), resulting in an average classification accuracy of ~99%.

### Informational content of single neural activity features

Another phase of our analysis examined the properties of individual neural activity features (threshold crossings and spike power for each recording channel). We used nearest neighbor classification in order to quantify the informational content of each feature independently. We evaluated classification accuracy within each of the six gesture groups separately, as well as the ability of individual neural features to discriminate between gesture groups (including "no action"), left vs. right hand, and flexion vs. extension movements. Overall, the percentage of features displaying classification accuracy above chance levels for at least one of the nine classification categories was ~57% for T11 and ~83% for T5. Individual features displayed unimodal, long-tailed distributions of classification accuracy values for each of the categories examined (Fig. 8). The range of classification accuracy distributions was similar across the two participants within the six gesture groups. However, participant T11 displayed higher single-feature classification accuracy than T5 for the flexion vs. extension category as well as classification between gesture groups, but lower values for left vs. right hand classification.

Finally, we examined the degree to which individual features combined different types of information. This was accomplished by counting the number of classification categories where a given feature displayed decoding accuracy above chance levels. These features were labeled as "selective". For both participants, the number of features ranked as selective for multiple categories decreased linearly with the number of categories considered (Fig. 9). Less than 20% of selective features displayed information related to a single gesture category, while more than half exceeded chance for at least five gesture categories.

### Topographical distribution of information

We examined decoding accuracy for individual electrodes (averaged across sessions) in order to assess the topographical distribution of information across the the four squared millimeters of cortical surface covered by each microelectrode array (note that since all electrodes were 1.5 mm long with a single recording site at the tip, it is not possible to evaluate the effect of recording depth). Features displaying the highest decoding accuracy for a given decoding category (e.g. discriminating between individual finger flexion movements) tended to originate from neighboring electrodes (Supplementary Figs. S4, S5). However, multiple non-contiguous 'hot spots' for each decoding category were identified across the cortical surface. Decoding hot spots for a given category evaluated for left or right gestures (e.g., four-way decoding of individual

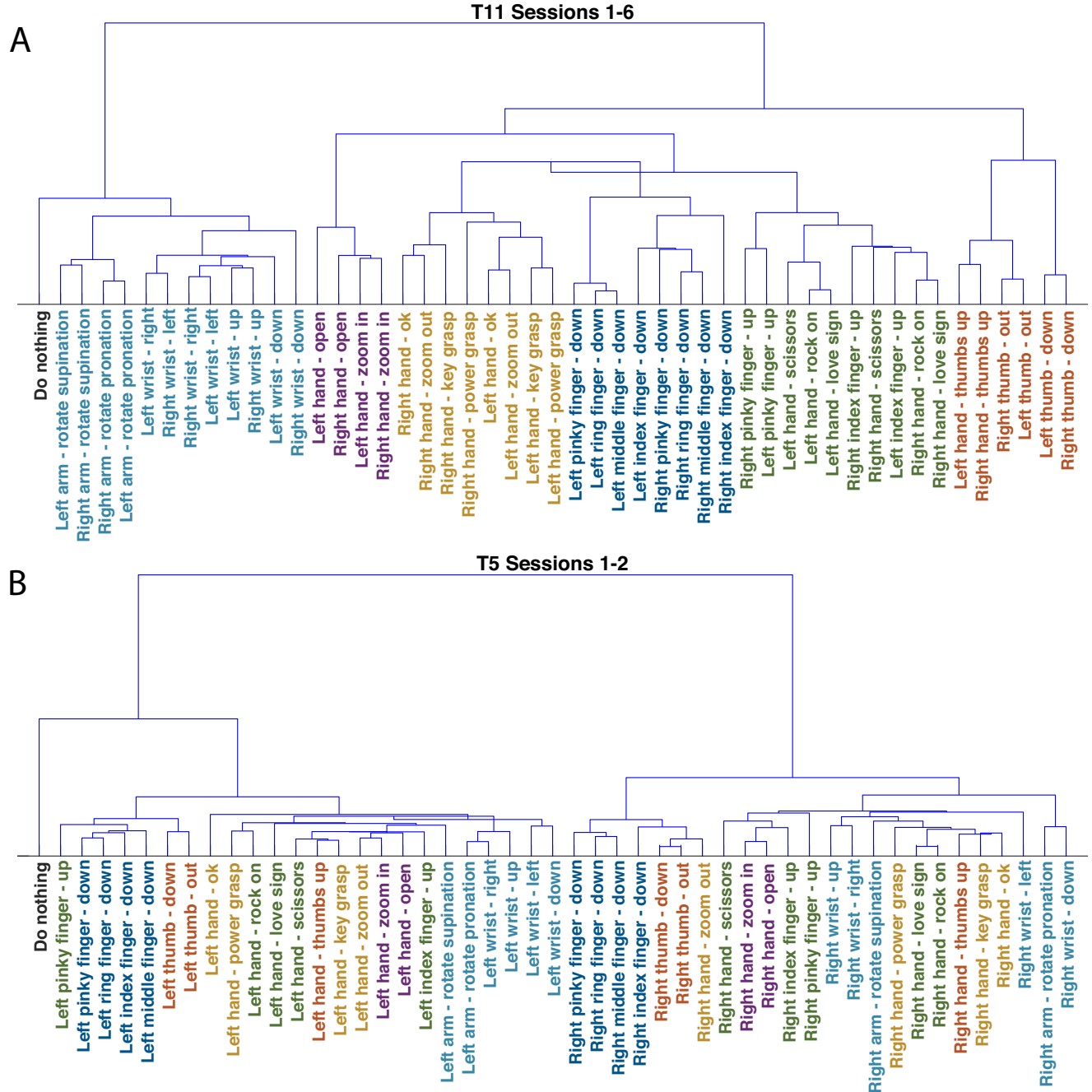

**Fig. 4 | Gesture dendrograms generated based on neural activity patterns.** Ward's method was used to generate dendrograms based on the relative position of gesture centroids in the unified latent spaces (combining all datasets for each participant). **A** Gesture dendrogram for participant T11. Colored according to gesture class (dendrogram branch). **B** Gesture dendrogram for participant T5, with colors corresponding to the same shape categories.

finger flexion on the right compared to the left hand) tended to overlap, suggesting that the same regions contributed to encoding movements independent of the effector. Two-way classification of effector (e.g., whether a gesture was performed with the right or left hand across all sets of gestures) presented a different pattern: higher decoding accuracy was observed on the ventral aspect of both microelectrode arrays for participant T5, displaying a more pronounced gradient across the cortical surface (Supplementary Fig. S6). Participant T11 also displayed this type of gradient, but with higher decoding accuracy towards the posterior medial aspect of the array. T11 also displayed a gradient with higher decoding accuracy in the anterior medial corner of the array for six-way classification of the identified gesture classes. A gesture class gradient was not evident for participant T5.

## Discussion

Our goal for this project was to examine the intrinsic structure of cortical neural features related to intended hand gestures in PCG. To our knowledge, the set of 49 discrete gestures investigated constitute the most detailed map of this kind of neural latent space in humans.

Our results show that intended gestures of both hands elicit highly specific neural activity patterns that can be readily discriminated using activity from a single hemisphere. We show that sets of approximately 10 gestures can be decoded with ~90% accuracy offline (Fig. 6). For participant T11, the dominant (right) hand (contralateral to the recording site) tended to display higher gesture discriminability than the non-dominant (left), however, overall the difference was relatively small (at most on the order of ~5% decoding accuracy, as shown in Fig. 7). Participant T5 presented nearly

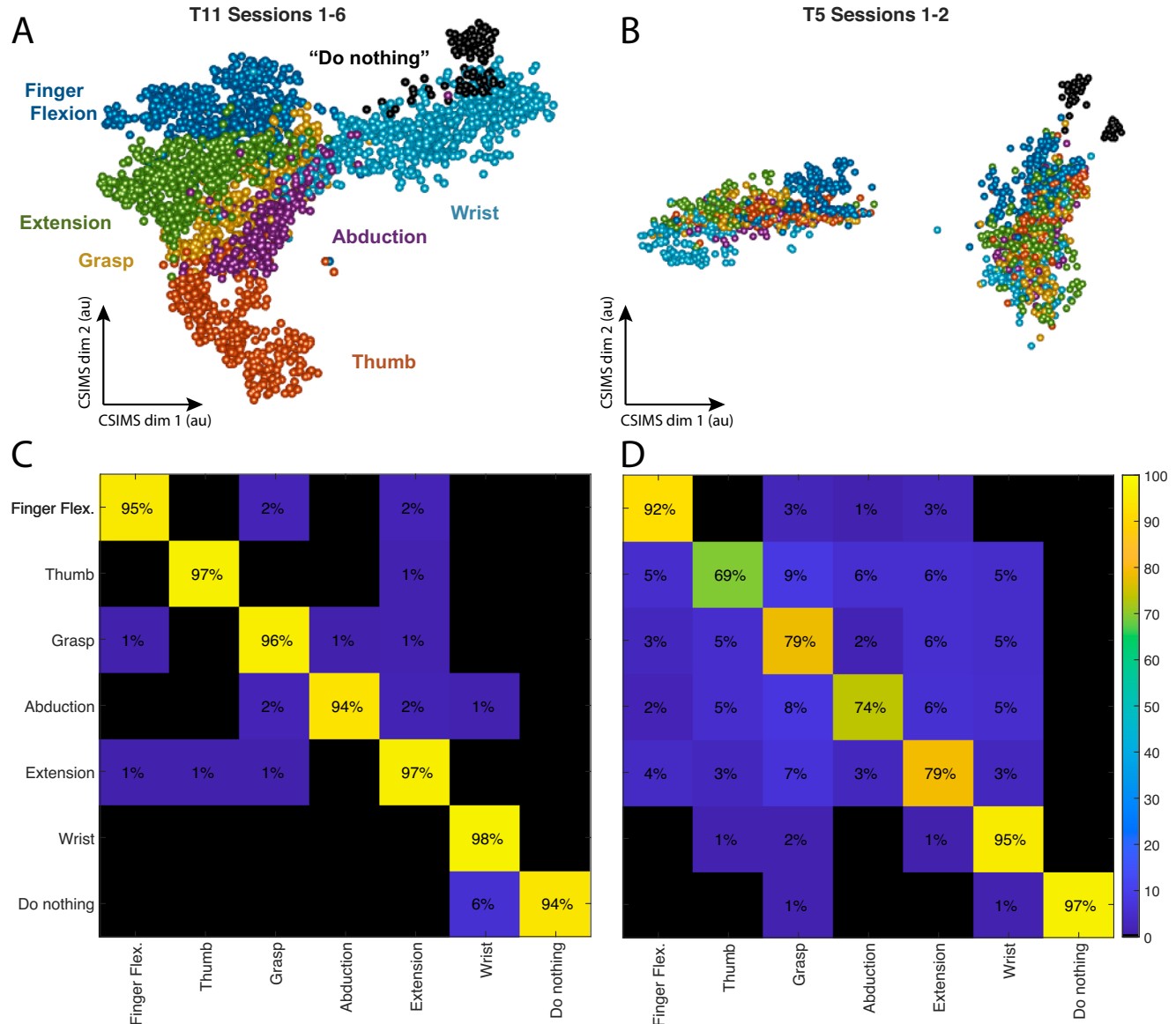

**Fig. 5 | Gesture latent spaces. A** 2D projection of the combined latent gesture space for T11, each point represents one of 7322 gestures across six sessions. Color denotes gesture groups shown in Fig. 4A. **B**. Latent gesture space for T5, representing 1906 gestures. Same color conventions as panel A. Note that for participant T5, gestures attempted with the right and left hand formed two distinct clusters (unlike T11, where the right and left version of each gesture were usually ranked as most similar). **C, D** Confusion matrices showing results from nearest neighbor classification, demonstrating the separation across the gesture groups for T11 and T5, respectively.

identical classification results for each hand. Our results underscore the prevalence of ipsilateral command signals in motor cortical areas which has been previously demonstrated in humans with paralysis and able-bodied monkeys[46–50], and demonstrate that this information includes subtle differences across a variety of gestures.

Despite the similarity in overall classification accuracy over the full set of gestures, the two participants displayed markedly different relative variance of neural features (assessed through dPCA, Fig. 3). For participant T11 ~50% total variance was associated with the gesture classes, while less than 5% was associated with the effector (right or left hand). By contrast, for participant T5 only ~22% of the variance was associated with gesture classes, and approximately one third of the variance was associated with the effector. Each participant also displayed a unique organization of their respective neural latent spaces (Fig. 5). For T11, the latent spaces for the right and left hand were closely aligned, with pairs of identical gestures performed by different hands being consistently ranked as most similar. By contrast, right and left hand latent spaces for T5 were almost completely decoupled, resulting in near perfect effector decoding.

The neural latent space for participant T11 displayed clear divisions between neural responses associated with six different groups of gestures (Fig. 5). For participant T5 two of the gesture categories (finger flexion and wrist movements) also formed independent clusters (Fig. 5). Although the distribution of neural responses for the four remaining categories displayed a greater degree of overlap, it was still possible to classify them well in excess of expected chance levels. Overall, classification errors across hands tended to be more common for T11, while errors across gesture classes were more common for T5. The informational content of individual neural features mirrored the results obtained from ensemble level analysis, with higher classification accuracy across gesture groups for T11, and more effector related information for T5 (Fig. 8). Overall, a higher percentage of features displayed gesture-related information in participant T5 (~83%) compared to T11 (~57%). The precise position of the microelectrode arrays on the cortical surface could contribute to the differences observed across participants: note that the topographic distribution of effector and gesture class classification tended to display relatively sharp boundaries (Supplementary Fig. S6). Another

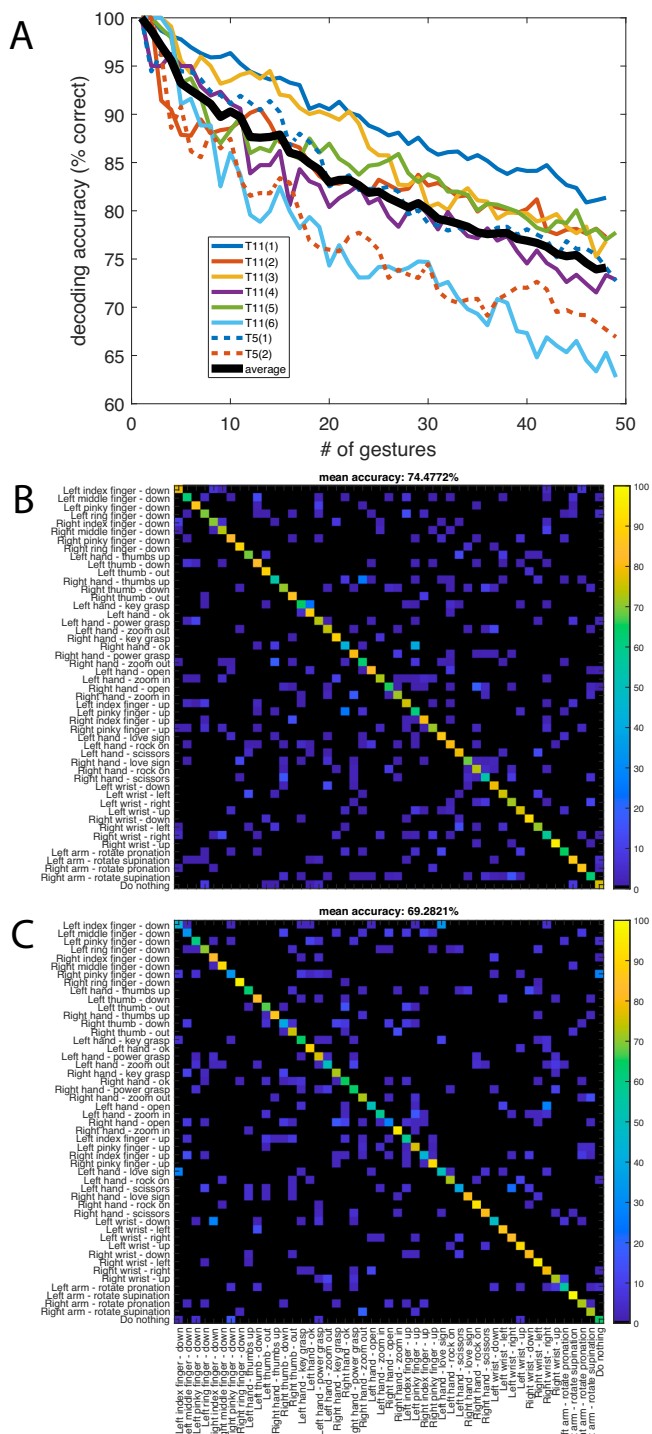

**Fig. 6 | Classification of 49 gesture classes. A** Overall gesture decoding accuracy as a function of number of gestures (data for each session shown separately, black line indicates average performance). **B** Full confusion matrix for T11 (six sessions combined within a single latent space). **C** Confusion matrix for T5 (two sessions).

individual could be reflected by the relationship between their effector-specific latent spaces.

## Holistic gesture encoding in the human precentral gyrus

These results suggest that the neural latent space of intended gestures reflects an information-rich signal stream that could potentially augment iBCI applications by providing additional intuitive and precise dimensions of control across both upper extremities, even when recorded from a single hemisphere. An orderly set of relationships between the neural ensemble activity patterns associated with different types of hand movements (Fig. 5) allowed for accurate classification across a wide set of gestures (Figs. 6,7). How is this information encoded across ensembles of individual neurons?

Previous work in monkeys has shown that the activity of single neurons (even cortico-motoneuronal cells) can change in a task-dependent manner and generally does not directly reflect EMG outputs[51–53]. For example, post-spike facilitation of the EMG for a specific muscle may be present for a precision grip, but not for a power grip engaging the same muscle to a greater extent[54]. It has been proposed that individual neurons may drive the coordinated activity of groups of muscles, sometimes referred to as muscle synergies[55–58]. Such synergies could explain the inconsistency between single neuron activity and EMGs if we consider that a given muscle may be part of multiple synergistic groups. In this case, we would only expect a given neuron to fire if a specific combination of muscles is to be activated, as is the case when performing hand gestures engaging multiple degrees of freedom. We could envision a system where a finite set of specific synergies are embodied by individual neurons. This would lead to distinct groups of neurons with stereotyped responses. Alternatively, it is possible that each neuron has a somewhat unique projection pattern towards downstream motor neuron pools, so that in practice the number of synergies approximates the number of neurons. Under this encoding scheme, we would expect heterogeneous responses among individual neurons whose activity reflects multiple, diverse aspects of the movement output. This holistic encoding scheme would embed specific types of information diffusely across large populations, rather than within small numbers of highly specialized, narrowly tuned neurons. Holistic encoding has been shown to combine proximal and distal movement parameters in monkeys[59]. Such 'mixed selectivity' has been proposed to provide computational benefits by supporting robust high-dimensional representations that facilitate the readout of a larger set of input-output functions by downstream neurons[60–62]. Holistic encoding is also compatible with 'compositional' representations where effector and movement information are encoded by independent dimensions of the neural response, as has been demonstrated for single unit activity related to whole-body attempted movement in the human precentral gyrus[50]. More generally, strategies used to study neural computation have shifted from a representational framework focusing on pairing external covariates (e.g., sensory cues, reaching/grasping movement kinematics) with single unit firing rates[63] to a more recent perspective, grounded in control theory, that views ensembles of cortical neurons as dynamical systems governed by low dimensional dynamics[44,64,65]. This view suggests that individual neurons will have heterogeneous, information-rich response properties reflecting multiple, potentially time-varying aspects of sensory-motor computations.

Our findings support the hypothesis of holistic gesture encoding in the human precentral gyrus. Decoding accuracy values for individual neural features revealed unimodal, long tailed distributions across all categories examined, including discriminating among related gestures (i.e., among individuated finger movements), gesture classes (i.e., prehension vs. extension movements), and effectors (i.e., right vs. left hand). The lack of multimodal peaks suggests a gradient of gesture-related modulation instead of a sharp divide between classes of neurons related to specific movement outputs. The observation that features tend to be informative with respect to multiple gesture categories (Fig. 9) highlights the rich informational content of intracortical signals. It is also worth noting that gestures engaging small numbers of joints (e.g., finger flexion) were decoded at similar levels of

factor that could influence the differences observed is the participants' mechanism and complexity of neural injury (T11 had a prior traumatic brain injury in addition to spinal cord injury, and T11 also reported being ambidextrous (though preferring his right hand more computer-based tasks) prior to his spinal cord injury; the two participants also had different AIS scores implying different levels and degree of spinal injury affecting both efferent and afferent activity: C4 AIS-C for T5 and C4 AIS-B for T11). It is also possible that the degree of ambidexterity for each

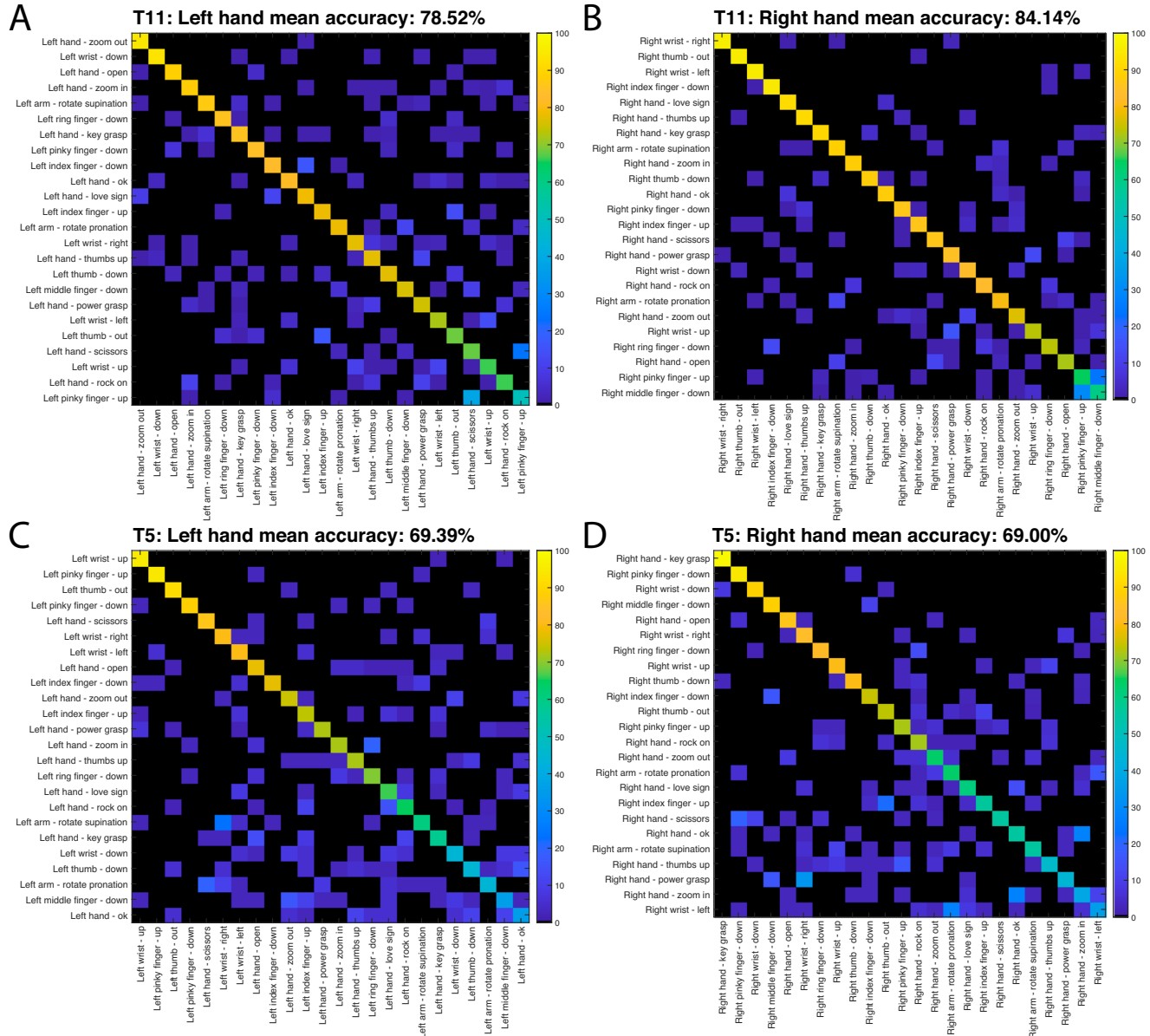

**Fig. 7 | Classifying right and left hand gestures separately.** Confusion matrices for each hand: T11 right (**A**), T11 left (**B**), T5 right (**C**), and T5 left (**D**). Nearest neighbor classification was performed with 10-fold cross-validation for each and separately. Gestures are arranged according to decoding accuracy for each hand.

accuracy as gestures engaging multiple joints (e.g., the "grasp" category) (Figs. 7,8). These findings agree with previous work in non-human primates showing that the movement of individual fingers is associated with widely overlapping populations of neurons, and single neurons are active for many different fingers[25,66–68]. This phenomenon can be partially explained by neural activity required to stabilize joints that are not intended to move, but nevertheless emphasizes the practical advantages of holistic encoding for the control of multi-joint systems with complex interdependencies, such as the hand. It is also remarkable that this encoding scheme is preserved in the neural activity patterns associated with movements attempted by people with tetraplegia.

Our results suggest that neural activity patterns elicited by intended hand gestures could provide a large number of discrete commands that could be made accessible for people using iBCI systems. While the present work was limited to examining intended movements engaging either the left or right hand independently, it is likely that bimanual movements (where both hands move simultaneously) would further extend the range of distinct activity patterns suitable for iBCI control.

## Methods
### Participants

All ethical regulations relevant to human research participants were followed. The research participants (identified as T11 and T5) gave informed consent to the study and publications resulting from the research. T11 is a right-handed man, 36 years old at the time of the study, with a history of tetraplegia secondary to a cervical spinal cord injury (C4 AIS-B SCI) that occurred 11 years prior and accompanied by some cerebral injury (full history unknown); 5 years before that event, he experienced a traumatic brain injury with a subdural hemorrhage requiring surgical evacuation, with full recovery. At the time of the sessions, T5 was a 68-year-old right-handed man with a C4 AIS-C SCI that occurred approximately 9 years prior to study enrollment. Both participants had two 96-channel intracortical microelectrode arrays placed chronically into the left PCG as part of the ongoing BrainGate pilot clinical trial (www.ClinicalTrials.gov; Identifier: NCT00912041). Details of the recording paradigm have been described previously[14,69–71]. Microelectrode array implantation targeted the "hand knob" area of the left precentral gyrus, which was identified by pre-operative

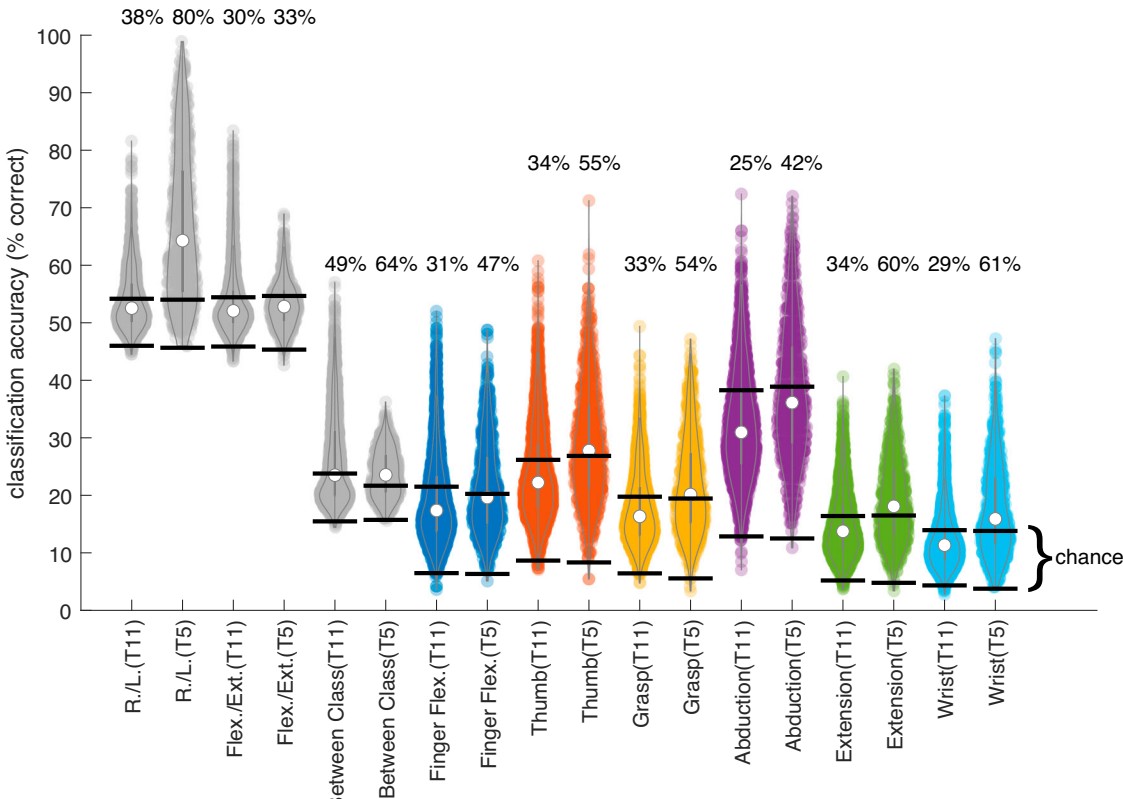

**Fig. 8 | Single neural activity feature classification accuracy.** Each violin plot shows the distribution of single-feature classification accuracy values across the gesture categories specified for one participant. The 95% confidence intervals of the chance distributions (calculated using 1000 random permutations of the gesture labels) are indicated by black lines, and the percentage of features exceeding the upper confidence boundary for the chance distribution are indicated above each plot.

magnetic resonance imaging (MRI). A description of the precise locations of the implants for T11 and T5 is provided in refs. 72, 73, respectively. This study was performed with permissions of U.S. Food and Drug Administration (Investigational Device Exemption #G090003; CAUTION: Investigational Device. Limited by Federal Law to Investigational Use) and the Institutional Review Boards of Massachusetts General Hospital, VA Providence Healthcare System, Brown University, and Stanford University. All research sessions were performed at the participant's place of residence.

**Gesture task implementation**

We reviewed the set of gestures with the participants before the first session, and demonstrated each one to help the participants understand the cues. Aside from this exposure, no additional training was provided. Each session began with 2–4 open-loop (OL) blocks, where the task proceeded independently of recorded neural activity. Each open-loop block included at least two trials with each of the set of 48 gestures. For OL blocks, the attempt line turned bright blue when the box intersected the attempt line, to emphasize the instructed time of movement initiation ("go" cue). OL blocks included the full set of gestures. Neural activity recorded during OL blocks was used to train decoders for subsequent closed-loop (CL) blocks. Closed-loop decoding was performed using multiclass LDA followed by a HMM[40] (as described below). CL blocks used sets of either six or seven gestures, with ~10 repetitions of each one. For CL blocks, feedback was provided to the participants based on neural activity in real time: when the cued gesture was decoded, the target and attempt line turned green; if any other gesture (except "do nothing") was decoded, both objects turned red instead. In order to discourage participants from "jumping the gun" (attempting the cued gesture before the go cue) the attempt line would turn yellow if the correct gesture was decoded before the target intersected with the attempt

line. This last task feature was added for the last three sessions with T11 and both sessions with T5. The duration of the delay period (i.e., the time between the appearance of the gesture cue and the time when it reached the attempt line) varied across sessions in order to accommodate the preferred trial speed for each participant. Specifically, the first 3 sessions for T11 had a 1 s instructed delay, which was increased to 2 s for the last three sessions. For participant T5 a 3 s delay was used.

**Statistics and reproducibility**

The reported results were replicated across six sessions for participant T11 and two sessions for participant T5. All sessions recorded with the participants for this task were included in the analysis. Statistical analysis focused on evaluating the informational content of signals for each subject as outlined below.

**Estimating the number of task-responsive features over time**

To gain an understanding of the task-related neural recruitment over time, we computed the distribution of features that were task-responsive to one or more conditions ($p < 0.001$, Wilcoxon Rank-Sum, Bonferroni corrected by class). We compared the time-averaged feature response from a baseline period (500 ms window starting −1500 ms prior to gesture cue onset to a sliding window (500 ms causal window from −2.3 s to 2.3 s around the attempt cue stepping by 20 ms increments).

**Closed-loop decoding with LDA**

Gestures were decoded with a multiclass LDA followed by a HMM[40]. There was one class per each gesture and an additional "no decode" class. Coefficients were estimated with a regularization term of 0.1 for the first three sessions of T11, and 0.45 for session 4–6 for T11 and for T5. The selected z-scored neural features were smoothed with a 100 ms boxcar window and

**Fig. 9 | Individual neural features convey information related to multiple gesture categories.** Orange bars show the percentage of selective features that displayed classification accuracy exceeding chance levels for exactly the number of categories indicated. Only features exceeding chance in at least one category were included in this analysis. Blue bars represent the percentage of features exceeding chance levels for at least the number of categories specified (cumulative sum).

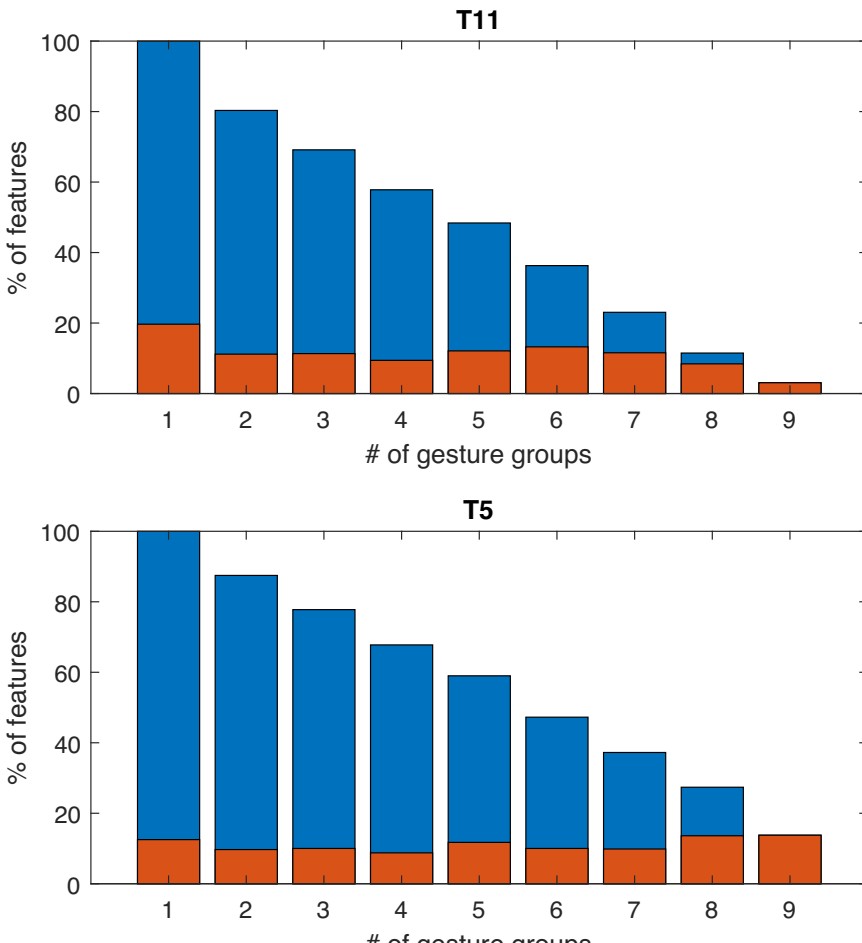

were projected into a LDA space. We learned the emission mean and covariance for the HMM from the empirical mean and covariance of the projected neural features from the open-loop blocks. We set the transition matrix using the proportion of transitions between gesture states and no action state. After the class probabilities were estimated, they were normalized with a softmax function and were smoothed with a 400 ms boxcar window. A gesture was decoded when a class probability was above the threshold of 95%.

## Demixed principal component analysis

Demixed principal component analysis was applied to find the set of components that best explained task-related variables[41]. Finding the demixed principal components involves decomposing the data using an ANOVA-like approach, where the goal is to separate out the marginalized contributions of time, gestures, and hand conditions, as well as their interactions. Then, through optimization, identifying the top components that best reflect the respective marginalizations.

To illustrate this, consider a single neural response over a time period $t$, with $g$ different gestures for each of the $h$ hand conditions. This data can be represented as a tensor of size $[t \times g \times h]$. The neural response can then be decomposed into its constituent components:

$$X_{tgh} = \bar{x} + \bar{x}_t + \bar{x}_g + \bar{x}_h + \bar{x}_{tg} + \bar{x}_{th} + \bar{x}_{tgh}$$

Where

$\bar{x}$ is the overall mean response,

$\bar{x}_t$, $\bar{x}_g$, $\bar{x}_h$ represent the contributions of time, gestures, and hand conditions, and

$\bar{x}_{tg}$, $\bar{x}_{th}$, $\bar{x}_{tgh}$ capture the interactions between these conditions.

And are estimated by

$$\bar{x} = <X_{tgh}>_{tgh}$$

$$\bar{x}_t = <X_{tgh} - \bar{x}>_{gh}$$

$$\bar{x}_g = <X_{tgh} - \bar{x} - \bar{x}_t>_{th}$$

$$\bar{x}_h = <X_{tgh} - \bar{x} - \bar{x}_t - \bar{x}_g>_{tg}$$

$$\bar{x}_{tg} = <X_{tgh} - \bar{x} - \bar{x}_t - \bar{x}_g - \bar{x}_h>_{h}$$

$$\bar{x}_{th} = <X_{tgh} - \bar{x} - \bar{x}_t - \bar{x}_g - \bar{x}_h - \bar{x}_{tg}>_{g}$$

$$\bar{x}_{tgh} = X_{tgh} - \bar{x} - \bar{x}_t - \bar{x}_g - \bar{x}_h - \bar{x}_{tg} - \bar{x}_{th}$$

Here, $<X>_{yz}$ denotes averaging X along the y and z dimensions. Note that while $<X_{tgh}>_{tgh}$ is a scalar value, all values above are tiled to maintain the original tensor size, $[t \times g \times h]$. We combine the components into the following marginalizations ($X_m$) of interest:

$$X_{common} = \bar{x}_t$$

https://doi.org/10.1038/s42003-025-08557-z **Article**

$$X_{gesture} = \bar{x}_g + \bar{x}_{tg}$$

$$X_{hand} = \bar{x}_h + \bar{x}_{th}$$

This process is repeated for each of the $N$ neural features, which are reshaped into an $N$ by $tgh$ matrix to estimate for the encoder and decoder matrices for each marginalization by minimizing the loss ($L_m$) for the following optimization problem.

$$L_m = \|X_m - E_m D_m X\|^2$$

Where $\| \|^2$ is the Frobinius norm, $E_m$ is the encoder matrix, and $D_m$ is the decoder matrix. Each row of $D_m$ yields one demixed principal component. Components are ordered according to the amount of variance explained. As in the original paper, we estimate the dPCs by combining neural features across multiple sessions. We applied regularization with a ridge penalty. The optimal regularization parameter was determined via cross-validation, using the default settings from the dPCA toolbox.

### Generating neural latent spaces using pairwise similarity metrics

We used pairwise similarity metrics combined with dimensionality reduction in order to generate low-dimensional latent spaces that would capture the intrinsic relationships between the activity patterns associated with specific hand gestures.

Each neural feature vector was smoothed using a 200 ms Gaussian kernel. A pairwise similarity matrix was calculated for each normalized feature using an euclidean distance metric. Only features displaying gesture-related information were included in the analysis (see 'Informational content of single features' section for further details). The matrices were then concatenated before applying dimensionality reduction using t-distributed Stochastic Neighbor Embedding (t-SNE)[74] initialized using principal component analysis. t-SNE is well-suited to similarity analysis because it explicitly attempts to preserve nearest-neighbor relationships in the data by minimizing KL-divergence between local neighborhood probability functions in the high and low-dimensional spaces. That is, it explicitly attempts to preserve relationships between data points that are close together in the high-dimensional space, which makes it ideal for analyzing neural datasets that lie on or close to a nonlinear manifold. PCA initialization helps to preserve global shape and inter-cluster relationships[75,76]. The non-random PCA initialization also ensures reproducibility across iterations. In the final low-dimensional "similarity space", a neural activity pattern is represented by a single point. The distance between points denotes the degree of similarity between the activity patterns they represent.

### Latent space alignment

Latent spaces were aligned across sessions by calculating a Procrustes transform (i.e., rotation, translation, reflection, and uniform scaling implemented through matrix multiplication), as described in ref. 45. The transform was calculated by minimizing the mean squared error between sets of matched landmarks. Here, we used the centroids of the gesture classes as landmarks. The transform computed using the centroids was then applied to each individual point in order to project them into a common coordinate space. Note that this shape-preserving transform preserves the intrinsic structure of each latent space.

### Reporting summary

Further information on research design is available in the Nature Portfolio Reporting Summary linked to this article.

### Data availability

All neural data needed to reproduce the findings in this study are publicly available at the Dryad repository[77]. The dataset contains neural activity features recorded while participants attempted a total of 8249 individual actions spanning 42 different gesture types.

### Code availability

Code that implements an offline reproduction of the central findings in this study is publicly available on GitHub at https://github.com/cvargasi/GestureEncoding.

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

## Acknowledgements

The authors would like to thank participants T5, T11, their families and carepartners, Beth Travers, Dave Rosler, Maryam Masood, Sandrin Kosasih, Beverly Davis, and Kathy Tsou for their contributions to this research. Research was supported by NIH-NIDCD (U01DC017844, R01DC014034), the Office of Research and Development, Rehabilitation Research Development and Translation, Dept of Veterans Affairs (A2295R, A2827R, A3803R, N2864C), NIH NIMH (T32MH115895), NIH-NINDS (UH2NS095548, U01NS123101), AHA (19CSLOI34780000), the Croucher Foundation, Larry and Pamela Garlick, Wu Tsai Neurosciences Institute, Howard Hughes Medical Institute (HHMI) at Stanford, Simons Foundation Collaboration on the Global Brain 543045, Hong Seh and Vivian W.M. Lim endowed chair and HHMI Investigatorship at Stanford University.

## Author contributions

C.V.I. conceived the study and wrote the first draft of the manuscript. T.S.C. and T.H. developed the task interface and decoding software. N.P.S. coordinated data collection with T5 and provided feedback on the task and data analysis. C.V.I., A.K., C.N., D.T.A., and F.K. collected the data. C.V.I., T.H., J.T.G.,. and T.K.P. performed data analysis. L.R.H. is the sponsor-investigator of the multi-site clinical trial. Z.M.W. planned and performed T11's array placement surgery. J.M.H. planned and performed T5's array placement surgery and was responsible for all clinical-trial-related activities at Stanford. C.V.I., L.R.H., J.D.S., and J.M.H. supervised and guided the study. All authors reviewed and edited the manuscript.

## Competing interests

The content is solely the responsibility of the authors and does not necessarily represent the official views of NIH or the Department of Veterans Affairs or the United States Government. The MGH Translational Research Center has a clinical research support agreement (CRSA) with Axoft, Neuralink, Neurobionics, Paradromics, Precision Neuro, Synchron, and Reach Neuro, for which LRH provides consultative input. L.R.H. is a non-compensated member of the Board of Directors of a nonprofit assistive communication device technology foundation (Speak Your Mind Foundation). Mass General Brigham (MGB) is convening the Implantable Brain-Computer Interface Collaborative Community (iBCI-CC); charitable gift agreements to MGB, including those received to date from Paradromics, Synchron, Precision Neuro, Neuralink, and Blackrock Neurotech, support the iBCI-CC, for which LRH provides effort. J.M.H. is a consultant for Neuralink and Paradromics, is a shareholder in Maplight Therapeutics and Enspire DBS, and is a co-founder and shareholder in Re-EmergeDBS. He is also an inventor on intellectual property licensed by Stanford University to Blackrock Neurotech and Neuralink. All other authors declare no competing interests.
