## [Transparent Peer Review file · Communications Biology]

Gesture Encoding in human left precentral gyrus neuronal ensembles

Corresponding Author: Dr Carlos Vargas-Irwin

Version 0:

Reviewer comments:

Reviewer #1

(Remarks to the Author)

The authors describe classification of 48 gestures and a rest condition with micro electrode arrays of two SCI patients. The experiments were run in open and close loop paradigms. Their method of dimensionality reduction with dPCA leads to interesting results. The results are successful classification of gestures but also reveals clear differences between the two participants. Interestingly, classification scores of ipsilateral gestures were high as well.

The manuscript gives a very good overview of the different approaches on methodology and highlights all different findings in the rich data.

The discussion on the differences between the participants is thorough and leaves very few questions open. The discussion on holistic gesture encoding is relevant, but could be shortened with reference to papers advocating this idea.

The experimental methods are meticulous and well described.

Further comments:

- 107: spiking activity (threshold crossings -> is this multi-unit or single-unit? How sure are you that you can concatenate or pool over sessions?
- 210-213 `please correct figure numbers
- Fig 7: T11 right (A), T11 left (B), T5 right (C) and T5 left (D): please add participant to the subfigure titles instead of in the legend
- 455 Out of curiosity: what is the authors' hypothesis on the inter-individual variability in healthy persons? Which of the patients likely represents the healthy situation better?
- 517: this seems in contradiction with fMRI results showing finger movements separated in the PCG
- 567: Did the participants train on the gestures before recordings?

Reviewer #2

(Remarks to the Author)

Vargas-Irwin and colleagues investigated gesture encoding in the human precentral gyrus (PCG) using intracranial recording data from two tetraplegic participants. The study involves an extensive dataset of 49 discrete hand gestures from both hands, with a detailed comparison of cortical encoding across electrodes and between participants. The results provide evidence supporting the feasibility of decoding gestures using single-unit ensemble activity from a single hemisphere. Overall, the study effectively links the generation of hand movements to ensemble activities of PCG neurons using a brain-computer interface (BCI) system.

Comment 1

This is an interesting physiological study addressing the neuronal mechanisms underlying hand motor control. However, it does not appear to be an engineering study focused on achieving high classification accuracies. If this interpretation is correct, the authors should clearly highlight the novelty of their study. After reading the manuscript, it seems that the novelty lies in the finding that hand movements are associated with widely overlapping neuronal populations, regardless of whether they are ipsilateral or contralateral to the recorded hemisphere, a discovery made for the first time in humans. Is this correct? If so, statements emphasizing engineering achievements could be toned down to avoid misunderstanding the study's novelty. For instance, the statement in the abstract, "it was possible to achieve classification accuracies of ~70% for all 48

gestures (and ~90% for sets of 10)," could be rephrased. Additionally, lines 528–532 and 536–539 could be omitted to streamline the manuscript. Otherwise, without comparison to existing benchmarks, the current BCI performance may be difficult to interpret as superior.

Comment 2

Could the design of closed-loop decoding, performed using groups of six or seven gestures per block, have any order effects? In other words, would this design yield different BCI performance and physiological findings (such as activity of individual neurons reflecting diverse aspects of the movement output) compared to a design where all 48 gestures are attempted in a random order?

Comment 3

What is the relationship between the spatial location of electrodes in the PCG and the richness of information regarding gesture decoding? For example, are electrodes (or neurons) used for decoding multiple gesture patterns located in specific areas of the PCG or at certain depths? Is there a difference in the spatial organization of neurons involved in decoding ipsilateral versus contralateral movements? Additionally, is there a specific relationship between the feature values used for decoding and the gesture movement patterns?

Comment 4

(Lines 583-588) "The duration of the delay period (i.e. the time between the appearance of the gesture cue and the time when it reached the attempt line) varied across sessions in order to accommodate the preferred trial speed for each participant. Specifically, the first 3 sessions for T11 had a one second instructed delay, which was increased to 2 seconds for the last three sessions. For participant T5 a 3 second delay was used."

Given this variation, wouldn't the time between the gesture cue and the attempt cue differ across sessions? This discrepancy might influence the analysis, particularly regarding neural activity just before the attempt cue. In Fig. 1C and Fig. 3, neural activity is observed even outside the gesture trial period, suggesting that neural activity around the attempt cue may be influenced by these timing differences. Could you explain if and how this was accounted for?

Comment 5

(Lines 258 and 621) The evaluation of cluster distances in the latent space, obtained through t-SNE dimensionality reduction, using Ward's method raises some concerns. As t-SNE is designed to emphasize local similarities (close points) while not accurately reflecting relationships between distant points due to its KL-Divergence loss function, measuring distances between clusters in the latent space may not be a reliable evaluation. Additionally, as t-SNE's loss function is non-convex, different initializations may lead to different results. Were these results consistent across multiple initializations? Could you also provide details regarding the procedure for generating the latent space using t-SNE, including the hyperparameters used?

Minor comments

(Line 61) Typo error: "intracortical iBCIs" should be "iBCIs"

(Line 127) Typo error: "Fig. 1B" should be "Fig. 1C"

(Figs. 2B&C) Please consider indicating in the figure that the vertical and horizontal axes correspond to Attempted and Decoded, respectively.

(Line 204) "... demixed principal component analysis (Kobak et al. 2016), see Methods for additional details).": No additional details are provided in the Methods section regarding dPCA to estimate information content and dimensionality.

(Lines 593-596) "We compared the time-averaged feature response from a baseline period (500ms window starting -1500ms prior to attempt cue) to a sliding window (500ms causal window from -2.3 sec to 2.3 seconds around the attempt cue stepping by 20 ms increments).": If the attempt cue occurs at time 0 as indicated in Fig. 1C, the baseline period (i.e., -1.5 to -1 s to the attempt cue) could overlap with the period of interest (i.e., -2.3 s to 2.3 s around the attempt cue). This overlap sounds inappropriate for paired comparisons and raises concern about why paired comparisons of the same data window yielded statistically significant differences, particularly in participant T5.

Reviewer #3

(Remarks to the Author)

Gesture Encoding in human left precentral gyrus neuronal ensembles
Vargas-Irwin et al.

GENERAL COMMENTS

In this study, the authors asked two participants to engage in 48 different hand gestures while they recorded intracortical activity using two Utah arrays. The manuscript centers on different analyses that describe the degree to which the different hand gestures can be inferred from different features derived from the signals detected by the intracortical arrays. The primary results show that the correct gesture can be inferred from the brain signals about 70% of the time, which is much better than the accuracy expected by chance.

This study was conducted by a very experienced team, and data collection and analysis were performed at a high level of rigor. It is also undisputed that such data in humans are extremely rare. Moreover, the manuscript is written clearly, and the figures are of high quality. In sum, this reviewer is satisfied with the technical aspects of this work, and cannot find any substantive technical issues that would need to be corrected.

This reviewer's major concern is focused on the contribution this study is making. It is true that, as the authors point out, no human study using intracortical recording has employed such a large number of gestures. However, it does not appear that this study is making, or even attempting to make, a contribution to our fundamental understanding of hand motor control in humans. From the Introduction, it appears that the authors are primarily placing their work in the context of reconstruction of function in people with paralysis. If that is so, how is the present study presenting an advance towards the eventual goal of brain-based movement control in the target population? There does not seem to be a hypothesis (e.g., that the present study can decode more/more accurate information, or do so more rapidly, compared the (now already relatively extensive) work on decoding of movements in humans using different types of brain signals). Without any clear hypothesis, corresponding comparative analyses relative to previous work, and a clear and realistic discussion of the implications of the authors' work, the impact of the present study, while undeniably sophisticated, is unfortunately unclear.

Version 1:

Reviewer comments:

Reviewer #1

(Remarks to the Author)

The authors have satisfactorily addressed my comments and I recommend publication.

Reviewer #2

(Remarks to the Author)

The authors addressed all of my comments in a well organized manner.

Reviewer #3

(Remarks to the Author)

Gesture Encoding in human left precentral gyrus neuronal ensembles
Vargas-Irwin et al.

In their revision, the authors produced additional analyses and corresponding text to clarify the purpose of the study and to add important additional information. With this increased clarity, some questions emerge.

First, individual gesture classification was done for all available data as described in the manuscript: "Note that this analysis was performed offline, combining data collected from all available open and closed loop control blocks." However, what data was used for the training and testing? Was there cross-validation of the results? The manuscript described the use of LDA for closed-loop decoding but not for the offline gesture classification, although Fig. 7 seems to suggest that there was a nearest neighbor classifier and 10-fold cross validation.

Second, the manuscript specifies that "for T11, cue onset (the appearance of the instructed gesture, prior to the 'go' cue at time zero) was either 1 or 2 seconds before the go cue." How do the authors explain that, in T11, neural signals begin to change about 2.5 seconds before the go cue (Fig. 3B) and that there is no indication in the results that the data was collected for either 1 or 2 seconds before the go cue (e.g., two peaks in the results in Fig. 3B-D)?

Third, in that vein, Fig. 1C shows that, in T11, neural signals begin to change 1.5 seconds before the go cue, which is clearly different than the time courses shown in Fig. 3. This reviewer understands that the analyses are different, but the underlying neural signals cannot have different time courses unless results for different datasets are shown here or there is some mistake.

Fourth, the subjects' task was to begin attempting movements when the falling box crossed the horizontal line, but there is no indication that the subjects followed this instruction as discriminable neural signals clearly begin much earlier than the go cue and classification results are best for data including the time before the go cue. How can the authors exclude the possibility that some of the results described here can be explained by visual information? Additional interpretation/description is needed.

Fifth, in general, were all analyses reported here done on all data? Fig. 2 talks about classification accuracy in 60 trials (one block) (Fig. 2B-C), but are data in Fig. 2A for all X sets of 6/7 gestures? Please make it clear in the text and the figure legends what data were used to produce the results.

Version 2:

Reviewer comments:

Reviewer #3

(Remarks to the Author)

The authors made important final revisions that satisfactorily addressed all my remaining concerns.

Response to reviewer's comments for: Gesture encoding in human left precentral gyrus neuronal ensembles

Carlos E. Vargas-Irwin^{1,2,3}, Tommy Hosman^{3,5}, Jacob T. Gusman^{4,5,2,3}, Tsam Kiu Pun^{4,5,2}, John D. Simeral^{3,5,2}, Tyler Singer-Clark⁶, Anastasia Kapitonava⁶, Claire Nicolas⁶, Nishal P. Shah⁷, Donald Avansino⁸, Foram Kamdar⁷, Ziv Williams⁹, Jaimie M. Henderson^{7,10,11}, Leigh R. Hochberg^{3,5,2,6}

1. Department of Neuroscience, Brown University, Providence, RI, USA
2. Robert J. and Nancy D. Carney Institute for Brain Science, Brown University, Providence, RI, USA
3. VA Center for Neurorestoration and Neurotechnology, VA Providence Healthcare System, Providence, RI, USA
4. Biomedical Engineering Graduate Program, School of Engineering, Brown University, Providence, RI, USA.
5. School of Engineering, Brown University, Providence, RI, USA
6. Center for Neurotechnology and Neurorecovery, Department of Neurology, Massachusetts General Hospital, Harvard Medical School, Boston, MA, USA
7. Department of Neurosurgery, Stanford University
8. Howard Hughes Medical Institute at Stanford University, Stanford, CA, USA
9. Department of Neurosurgery, Massachusetts General Hospital, Harvard Medical School, Boston, Massachusetts 02114, USA.
10. Wu Tsai Neurosciences Institute, Stanford University, Stanford, CA, USA
11. Bio-X Institute, Stanford University, Stanford, CA, USA

Corresponding author email: Carlos_Vargas_Irwin@Brown.edu

We thank the reviewer for their feedback and believe the manuscript has benefitted from their suggestions. Specific reviewer comments are shown in blue, with the response provided below. Modifications to the original manuscript are highlighted. Comments and responses are grouped thematically.

Reviewer #1:

The authors describe classification of 48 gestures and a rest condition with micro electrode arrays of two SCI patients. The experiments were run in open and close loop paradigms. Their method of dimensionality reduction with dPCA leads to interesting results. The results are successful classification of gestures but also reveals clear differences between the two participants. Interestingly, classification scores of ipsilateral gestures were high as well. The manuscript gives a very good overview of the different approaches on methodology and highlights all different findings in the rich data.

The discussion on the differences between the participants is thorough and leaves very few questions open. The discussion on holistic gesture encoding is relevant, but could be shortened with reference to papers advocating this idea.

The experimental methods are meticulous and well described.

Reviewer #2:

Comment 1:

This is an interesting physiological study addressing the neuronal mechanisms underlying hand motor control. However, it does not appear to be an engineering study focused on achieving high classification accuracies. If this interpretation is correct, the authors should clearly highlight the novelty of their study. After reading the manuscript, it seems that the novelty lies in the finding that hand movements are associated with widely overlapping neuronal populations, regardless of whether they are ipsilateral or contralateral to the recorded hemisphere, a discovery made for the first time in humans. Is this correct? If so, statements emphasizing engineering achievements could be toned down to avoid misunderstanding the study's novelty. For instance, the statement in the abstract, "it was possible to achieve classification accuracies of ~70% for all 48 gestures (and ~90% for sets of 10)," could be rephrased. Additionally, lines 528–532 and 536–539 could be omitted to streamline the manuscript. Otherwise, without comparison to existing benchmarks, the current BCI performance may be difficult to interpret as superior.

Reviewer #3:

In this study, the authors asked two participants to engage in 48 different hand gestures while they recorded intracortical activity using two Utah arrays. The manuscript centers on different analyses that describe the degree to which the different hand gestures can be inferred from different features derived from the signals detected by the intracortical arrays. The primary results show that the correct gesture can be inferred from the brain signals about 70% of the time, which is much better than the accuracy expected by chance.

This study was conducted by a very experienced team, and data collection and analysis were performed at a high level of rigor. It is also undisputed that such data in humans are extremely rare. Moreover, the manuscript is written clearly, and the figures are of high quality. In sum, this reviewer is satisfied with the technical aspects of this work, and cannot find any substantive technical issues that would need to be corrected.

This reviewer's major concern is focused on the contribution this study is making. It is true that, as the authors point out, no human study using intracortical recording has employed such a large number of gestures. However, it does not appear that this study is making, or even attempting to make, a contribution to our fundamental understanding of hand motor control in humans. From the Introduction, it appears that the authors are primarily placing their work in the context of reconstruction of function in people with paralysis. If that is so, how is the present study presenting an advance towards the eventual goal of brain-based movement control in the target population? There does not seem to be a hypothesis (e.g., that the present study can decode more/more accurate information, or do so more rapidly, compared the (now already relatively extensive) work on decoding of movements in humans using different types of brain signals). Without any clear hypothesis, corresponding comparative analyses relative to previous work, and a clear and realistic discussion of the implications of the authors' work, the impact of the present study, while undeniably sophisticated, is unfortunately unclear.

We thank the reviewers for their valuable guidance to better place the current research in the context of both its intrinsic hypothesis and its contribution to BCI research. Our fundamental hypothesis is that cortical neuronal ensembles in the precentral gyrus, even when as sparsely sampled as is possible with planar intracortical arrays, are active and uniquely engaged during a large number of intended hand gestures. The study presents by far the largest number of attempted gestures ever studied (48 vs 12), revealing that a wide range of subtly different gestures can be reliably decoded from even a few dozen cortical neurons. Our findings underscore the high informational content related to gestures at the level of neuronal ensembles in the precentral gyrus, providing a clear direction for the development of assistive BCI technology. We hope this study will set a benchmark for gesture classification accuracy in humans, and therefore believe it is important to highlight the exact figures obtained. However, we believe that the study goes beyond demonstrating that it is possible to decode large numbers of intended gestures of both hands from a relatively small sample of neurons (i.e. less than one hundred out of tens of millions) from a single hemisphere.

We used a data-driven approach to identify a potentially novel gesture taxonomy based solely on the intrinsic relationships between neural activity patterns in human motor precentral gyrus. It was possible to separate these gesture categories above chance levels in both participants. Nevertheless, the latent space representation of gestures varied across subjects suggesting that each person may have a unique encoding pattern (although handedness and precise recording location may also play a role). Although the limited number of subjects makes interpretation challenging, we are providing a novel approach and benchmark to organize gesture related information for human motor cortex microelectrode array data. We also examined how gesture information is reflected at the level of individual neural features. Our findings revealed that information is widely distributed among a neural population with heterogeneous responses. While this finding supports theories of holistic encoding and mixed selectivity, it does not support the hypothesis that gestures are encoded using a relatively small number of underlying synergies. We have added the following sections to the introduction in order to more clearly outline the goals of the paper:

Line 52>> In this study we aim to examine how information related to discrete hand gestures is encoded by neuronal ensembles in the human PCG and evaluate the potential use of these signals for intracortical brain-computer interface (iBCI) applications.

Line 69>> Although all these results highlight the high informational content of cortical areas related to dexterous hand movements, so far the neural representation of only a small subset of these actions has been explored. This restricted scope not only limits the practical applications of gesture decoding but also hinders our understanding of the broader neural mechanisms underlying dexterous hand movements. There is a need to explore the neural representation of the cortical area related to a wider range of dexterous hand gestures, which could potentially be harnessed for a more versatile iBCI system.

Line 80>> In addition to providing a benchmark for gesture classification accuracy using human intracortical signals, this study also introduces a novel way to organize gestures into functional groups based on the intrinsic similarity of their associated neural activity patterns. This bottom-up approach aims to map the intrinsic structure of the neural latent space for gestures, and examine whether latent space geometry is conserved across study participants. Our results reveal subject-specific organization that nevertheless displays similar informational content widely distributed among individual neural activity features with heterogeneous encoding properties reflecting a large number of potential movement synergies. These findings underscore the potential of intended gestures across both limbs as control signals for BCI applications, even when neural recordings are constrained to a single hemisphere.

Regarding lines 528–532 and 536–539: We agree that these sections do not add a new perspective beyond what is presented in the rest of the discussion, and have removed them as suggested by reviewer #2.

Reviewer #2 Comment 2

Could the design of closed-loop decoding, performed using groups of six or seven gestures per block, have any order effects? In other words, would this design yield different BCI performance and physiological findings (such as activity of individual neurons reflecting diverse aspects of the movement output) compared to a design where all 48 gestures are attempted in a random order?

To our knowledge this is the first study to examine large numbers of gestures (i.e. more than 12) in human motor cortex. We therefore were not sure what decoding accuracy it would be possible to achieve when designing the experimental sessions. We chose to present sets of seven gestures for closed loop decoding in order to achieve high decoding accuracy that would keep participants motivated. However, each session began with at least 3 open loop blocks where the 48 gestures were presented in a random order. This was not explicitly mentioned in the original text, we have added the text below to clarify this point. We did not observe differences in decoding accuracy between the open and closed loop blocks.

Line 98>> Each open loop block included 2 repetitions of each of the 48 gesture types, randomly intermixed. At least three such blocks were included at the beginning of each session. Data collected during these initial open loop blocks was used to train the decoders used for closed loop control.

Reviewer #2: Comment 3

What is the relationship between the spatial location of electrodes in the PCG and the richness of information regarding gesture decoding? For example, are electrodes (or neurons) used for

decoding multiple gesture patterns located in specific areas of the PCG or at certain depths? Is there a difference in the spatial organization of neurons involved in decoding ipsilateral versus contralateral movements? Additionally, is there a specific relationship between the feature values used for decoding and the gesture movement patterns?

We have added a new section in the results as well as three supplementary figures in order to address the questions raised by the reviewer. We have also added additional text in the discussion related to these findings. Note that the microelectrode array has no capability of recording multiple depth locations, so it is not possible to evaluate the effect of recording depth.

Line 365>> Topographical distribution of information

We examined decoding accuracy for individual electrodes (averaged across sessions) in order to assess the topographical distribution of information across the the four squared millimeters of cortical surface covered by each microelectrode array.(note that since all electrodes were 1mm long with a single recording site at the tip, it is not possible to evaluate the effect of recording depth). Features displaying the highest decoding accuracy for a given decoding category (e.g. discriminating between individual finger flexion movements) tended to originate from neighboring electrodes (Supplemental figures S4, S5). However, multiple non-contiguous 'hot spots' for each decoding category were identified across the cortical surface. Decoding hot spots for a given category evaluated for left or right gestures (e.g. four-way decoding of individual finger flexion on the right compared to the left hand) tended to overlap, suggesting that the same regions contributed to encoding movements independent of the effector. Two way classification of effector (e.g. whether a gesture was performed with the right or left hand across all sets of gestures) presented a different pattern: higher decoding accuracy was observed on the ventral aspect of both microelectrode arrays for participant T5, displaying a more pronounced gradient across the cortical surface (Supplemental figures S6). Participant T11 also displayed this type of gradient, but with higher decoding accuracy towards the posterior medial aspect of the array. T11 also displayed a gradient with higher decoding accuracy in the anterior medial corner of the array for six-way classification of the identified gesture classes. A gesture class gradient was not evident for participant T5.

Line 427>>The precise position of the microelectrode arrays on the cortical surface could contribute to the differences observed across participants: note that the topographic distribution of effector and gesture class classification tended to display relatively sharp boundaries (Supplemental figure S6). Another factor that could influence the differences observed is the participants' mechanism and complexity of neural injury (T11 had a prior traumatic brain injury in addition to spinal cord injury, and T11 also reported being ambidextrous (though preferring his right hand more computer-based tasks) prior to his spinal cord injury; the two participants also had

different AIS scores implying different levels and degree of spinal injury affecting both efferent and afferent activity: C4 AIS-C for T5 and C4 AIS-B for T11).

Newly added supplemental figures are included below, as specified in the Communications Biology Revised manuscript submission file checklist.

Figure S4. Finger Flexion decoding accuracy across the cortical surface. Each grid partition represents the position of a single electrode on a 10 by 10 grid (note that the corners are not connected, for a total of 96 active electrodes per array). Each square is colored according to the highest decoding accuracy for features recorded at that location (either threshold crossings or spike power) averaged across all sessions. Results for 4 way classification of individual finger flexion movements are shown separately for the left and right hands (left and right columns, respectively). Only one array is shown for participant T11 (top row), since the other array did not display

features with significant decoding. Two arrays (anterior and posterior) are shown for participant T5 (middle and bottom rows, respectively).

Figure S5. Flexion vs. Extension decoding accuracy across the cortical surface. Similar layout to figure S4, but showing 2-way classification of intended movements involving flexion vs. extension of the digits.

Figure S6. Gesture group / effector decoding accuracy across the cortical surface. Similar layout to figure S4, but showing 6-way classification of intended movements according to gesture class (left column) or 2-way classification of effector (i.e. right or left hand, right column).

Reviewer #2: Comment 4

Given this variation, wouldn't the time between the gesture cue and the attempt cue differ across sessions? This discrepancy might influence the analysis, particularly regarding neural activity just before the attempt cue. In Fig. 1C and Fig. 3, neural activity is observed even outside the gesture trial period, suggesting that neural activity around the attempt cue may be influenced by these timing differences. Could you explain if and how this was accounted for?

As the recording sessions progressed, we adjusted the timing of cues in order to maximize participant comfort, which was necessary to collect the large amount of samples needed to accurately represent neural activity patterns associated with 48 different gestures. This did result in a different duration for the delay period across sessions. In order to minimize the effect of the varying delay period, we centered our analysis window on the attempt cue. We also chose these windows based on the neural recruitment curves shown in figure 1, aiming to capture a time window when the neuronal populations were maximally engaged. In all cases, only data recorded after the gesture cue was presented was taken into account for analysis.

Reviewer #2: Comment 5

(Lines 258 and 621) The evaluation of cluster distances in the latent space, obtained through t-SNE dimensionality reduction, using Ward's method raises some concerns. As t-SNE is designed to emphasize local similarities (close points) while not accurately reflecting relationships between distant points due to its KL-Divergence loss function, measuring distances between clusters in the latent space may not be a reliable evaluation. Additionally, as t-SNE's loss function is non-convex, different initializations may lead to different results. Were these results consistent across multiple initializations? Could you also provide details regarding the procedure for generating the latent space using t-SNE, including the hyperparameters used?

We have expanded the description of our implementation of t-SNE providing additional details that address the reviewer's concerns. Specifically we initialized t-SNE using PCA as described below:

Line 638 >> The matrices were then concatenated before applying dimensionality reduction using t-distributed Stochastic Neighbor Embedding (t-SNE) (Hinton and van der Maaten 2008) initialized using principal component analysis. t-SNE is well suited to similarity analysis, because it explicitly attempts to preserve nearest-neighbor relationships in the data by minimizing KL-divergence between local neighborhood probability functions in the high and low dimensional spaces. That is, it explicitly attempts to preserve relationships between data points that are close together in the high-dimensional space, which makes it ideal for analyzing neural datasets that lie on or close to a nonlinear manifold. PCA initialization helps to preserve global shape and inter-cluster relationships (Kobak and Linderman, 2021; Lee et al., 2015). The non-random PCA-initialization also ensures reproducibility across iterations. In the final low-dimensional "similarity space", a neural activity pattern is represented by a single point. The distance between points denotes the degree of similarity between the activity patterns they represent.

(Line 204) "... demixed principal component analysis (Kobak et al. 2016), see Methods for additional details.": No additional details are provided in the Methods section regarding dPCA to estimate information content and dimensionality.

We added the following paragraph to the methods section providing more details on dPCA implementation:

Line 584>> Demixed Principal Component Analysis

Demixed principal component analysis was applied to find the set of components that best explained task related variables (Kobak et al., 2016)

[<https://github.com/machenslab/dPCA>]. Finding the demixed principal components involves decomposing the data using an ANOVA-like approach, where the goal is to separate out the marginalized contributions of time, gestures, and hand conditions, as well as their interactions. Then, through optimization, identifying the top components that best reflect the respective marginalizations.

To illustrate this, consider a single neural response over a time period t , with g different gestures for each of the h hand conditions. This data can be represented as a tensor of size $[t \times g \times h]$. The neural response can then be decomposed into its constituent components:

$$X_{tgh} = \bar{x} + \bar{x}_t + \bar{x}_g + \bar{x}_h + \bar{x}_{tg} + \bar{x}_{th} + \bar{x}_{tgh}$$

Where

\bar{x} is the overall mean response,

$\bar{x}_t, \bar{x}_g, \bar{x}_h$ represent the contributions of time, gestures, and hand conditions, and

$\bar{x}_{tg}, \bar{x}_{th}, \bar{x}_{tgh}$ capture the interactions between these conditions.

And are estimated by

$$\bar{x} = \langle X_{tgh} \rangle_{tgh}$$

$$\bar{x}_t = \langle X_{tgh} - \bar{x} \rangle_{gh}$$

$$\bar{x}_g = \langle X_{tgh} - \bar{x} - \bar{x}_t \rangle_{th}$$

$$\bar{x}_h = \langle X_{tgh} - \bar{x} - \bar{x}_t - \bar{x}_g \rangle_{tg}$$

$$\bar{x}_{tg} = \langle X_{tgh} - \bar{x} - \bar{x}_t - \bar{x}_g - \bar{x}_h \rangle_h$$

$$\bar{x}_{th} = \langle X_{tgh} - \bar{x} - \bar{x}_t - \bar{x}_g - \bar{x}_h - \bar{x}_{tg} \rangle_g$$

$$\bar{x}_{tgh} = X_{tgh} - \bar{x} - \bar{x}_t - \bar{x}_g - \bar{x}_h - \bar{x}_{tg} - \bar{x}_{th}$$

Here, $\langle X \rangle_{yz}$ denotes averaging X along the y and z dimensions. Note that while $\langle X \rangle_{tgh}$ is a scalar value, all values above are tiled to maintain the original tensor size, $[t \times g \times h]$. We combine the components into the following marginalizations (X_m) of interest:

$$X_{common} = \bar{x}_t$$

$$X_{gesture} = \bar{x}_g + \bar{x}_{tg}$$

$$X_{hand} = \bar{x}_h + \bar{x}_{th}$$

This process is repeated for each of the N neural features, which are reshaped into an N by tgh matrix to estimate for the encoder and decoder matrices for each marginalization by minimizing the loss (L_m) for the following optimization problem.

$$L_m = \|X_m - E_m D_m X\|^2$$

Where $\|\cdot\|^2$ is the Frobenius norm, E_m is the encoder matrix, and D_m is the decoder matrix. Each row of D_m yields one demixed principal component. Components are ordered according to the amount of variance explained. As in the original paper, we estimate the dPCs by combining neural features across multiple sessions. We applied regularization with a ridge penalty. The optimal regularization parameter was determined via cross-validation, using the default settings from the dPCA toolbox.

(Lines 593-596) “We compared the time-averaged feature response from a baseline period (500ms window starting -1500ms prior to attempt cue) to a sliding window (500ms causal window from -2.3 sec to 2.3 seconds around the attempt cue stepping by 20 ms increments).”: If the attempt cue occurs at time 0 as indicated in Fig. 1C, the baseline period (i.e., -1.5 to -1 s to the attempt cue) could overlap with the period of interest (i.e., -2.3 s to 2.3 s around the attempt cue). This overlap sounds inappropriate for paired comparisons and raises concern about why paired comparisons of the same data window yielded statistically significant differences, particularly in participant T5.

There was an error in the original description. The time period used as a baseline was taken prior to the gesture cue onset, not the attempt cue onset. The text has been corrected as shown below. We thank the reviewer for pointing this out and apologize for the confusion.

562>> Estimating the number of task responsive features over time

To gain an understanding of the task related neural recruitment over time, we computed the distribution of features that were task-responsive to one or more conditions ($p < 0.001$, Wilcoxon

Rank-Sum, Bonferroni corrected by class). We compared the time-averaged feature response from a baseline period (500ms window starting -1500ms prior to **gesture cue onset**) to a sliding window (500ms causal window from -2.3 sec to 2.3 seconds around the attempt cue stepping by 20 ms increments).

Reviewer #1:

- 107: spiking activity (threshold crossings -> is this multi-unit or single-unit? How sure are you that you can concatenate or pool over sessions?)

Given the thresholding method used, these features represented a mix of single and multiunit activity, varying channel by channel. Previous work has shown mixed evidence regarding the benefits of spike sorting for neural decoding (Christie et al. 2015; Todorova et al. 2014), in our experience it does not provide an immediate benefit and adds considerable computational burdens to the operation of a real-time system.

Data was pooled over sessions using ensemble-level latent space alignment. Each session was mapped into a common latent space using a shape preserving linear transform minimizing the distance between gesture cluster centroids (Procrustes method). For this method, it is not necessary to assume that channel identity would remain consistent across sessions.

Reviewer #1:

- 517: this seems in contradiction with fMRI results showing finger movements separated in the PCG

Original text:

517>> “These findings agree with previous work in non-human primates showing that the movement of individual fingers is associated with widely overlapping populations of neurons, and single neurons are active for many different fingers (Schieber and Hibbard 1993).”

We believe our findings are supported by previous fMRI as well as single unit studies. For example, Ejaz et al. (2015) describe activity associated with individual finger movements as follows based on fMRI:

“ Figure 1 shows a surface representation of activity patterns in M1 for three individual subjects (see Supplementary Fig. 1 for equivalent maps in S1). As reported earlier 5,6, there was no clear spatial segregation of finger activation patches. Instead, individual voxels were activated to varying degrees by all fingers, consistent with previous electrophysiological recordings that found that individual neurons have similarly broad tuning functions for finger movements 2,3.”

We have added additional citations to support and contextualize the statement as follows:

Line 496>> These findings agree with previous work in non-human primates showing that the movement of individual fingers is associated with widely overlapping populations of neurons,

and single neurons are active for many different fingers (Schieber and Hibbard 1993; Schieber 2002; Indovina and Sanes 2001; Ejaz et al. 2015).

With the following citations: Schieber, M.H. & Hibbard, L.S. How somatotopic is the motor cortex hand area? *Science* 261, 489–492 (1993).

Schieber, M.H. Motor cortex and the distributed anatomy of finger movements. *Adv. Exp. Med. Biol.* 508, 411–416 (2002).

Indovina, I. & Sanes, J.N. On somatotopic representation centers for finger movements in human primary motor cortex and supplementary motor area. *Neuroimage* 13, 1027–1034 (2001).

Ejaz, Naveed, Masashi Hamada, and Jörn Diedrichsen. 2015. “Hand Use Predicts the Structure of Representations in Sensorimotor Cortex.” *Nature Neuroscience* 18 (7): 1034–40.

Reviewer #1:

- 567: Did the participants train on the gestures before recordings?

We have added the following text to clarify how the gestures were presented to the participants:

537>> We reviewed the set of gestures with the participants before the first session, and demonstrated each one to help the participants understand the cues. Aside from this exposure, no additional training was provided. Each session began with 2-4 open loop (OL) blocks, where the task proceeded independently of recorded neural activity. Each open loop block included at least two trials with each of the set of 48 gestures.

Reviewer #1:

- 455 Out of curiosity: what is the authors' hypothesis on the inter-individual variability in healthy persons? Which of the patients likely represents the healthy situation better?

Given the small number of subjects (due to difficulty of obtaining intracortical single unit data from human participants), it is difficult to determine what trends would be present in a larger population. We believe that the precise location of the arrays is a likely contributor to the observed patterns. If arrays from both hemispheres were included, we expect that the signals would display both the clear separation between effectors (right vs. left) evident in participant T5, potentially accompanied by the precise separation between gesture classes observed for participant T11.

Reviewer #1:

- 210-213 `please correct figure numbers

We thank the reviewer for the correction, we have updated the manuscript to refer to figure 3 instead of 2.

191>> Here we find components that capture the response variance that is common across conditions (Common, Fig. 3 B, F), components that capture variance related to the attempted gesture (Gesture, Fig. 3 C, G), and components that capture variance related to the hand used (Hand/Lateral, Fig. 3 D, H).

Reviewer #1:

- Fig 7: T11 right (A), T11 left (B), T5 right (C) and T5 left (D): please add participant to the subfigure titles instead of in the legend

We have added the participant ID to each figure panel following the reviewer's suggestion.

Reviewer #2: (Line 61) Typo error: "intracortical iBCIs" should be "iBCIs"

We thank the reviewer for noticing this typo, it has been corrected.

Reviewer #2:(Line 127) Typo error: "Fig. 1B" should be "Fig. 1C"

We thank the reviewer for noticing this typo, it has been corrected.

Reviewer #2: (Figs. 2B&C) Please consider indicating in the figure that the vertical and horizontal axes correspond to Attempted and Decoded, respectively.

We have followed the reviewer's suggestion and modified the figure accordingly.

Response to reviewer's comments for: Gesture encoding in human left precentral gyrus neuronal ensembles

Carlos E. Vargas-Irwin^{1,2,3}, Tommy Hosman^{3,5}, Jacob T. Gusman^{4,5,2,3}, Tsam Kiu Pun^{4,5,2}, John D. Simeral^{3,5,2}, Tyler Singer-Clark⁶, Anastasia Kapitonava⁶, Claire Nicolas⁶, Nishal P. Shah⁷, Donald Avansino⁸, Foram Kamdar⁷, Ziv Williams⁹, Jaimie M. Henderson^{7,10,11}, Leigh R. Hochberg^{3,5,2,6}

1. Department of Neuroscience, Brown University, Providence, RI, USA
2. Robert J. and Nancy D. Carney Institute for Brain Science, Brown University, Providence, RI, USA
3. VA Center for Neurorestoration and Neurotechnology, VA Providence Healthcare System, Providence, RI, USA
4. Biomedical Engineering Graduate Program, School of Engineering, Brown University, Providence, RI, USA.
5. School of Engineering, Brown University, Providence, RI, USA
6. Center for Neurotechnology and Neurorecovery, Department of Neurology, Massachusetts General Hospital, Harvard Medical School, Boston, MA, USA
7. Department of Neurosurgery, Stanford University
8. Howard Hughes Medical Institute at Stanford University, Stanford, CA, USA
9. Department of Neurosurgery, Massachusetts General Hospital, Harvard Medical School, Boston, Massachusetts 02114, USA.
10. Wu Tsai Neurosciences Institute, Stanford University, Stanford, CA, USA
11. Bio-X Institute, Stanford University, Stanford, CA, USA

Corresponding author email: Carlos_Vargas_Irwin@Brown.edu

We thank the reviewer for their feedback, and have added additional text to expand on specific points that were not adequately described in the previous version. We believe these additions have improved the clarity of the manuscript, and hope that they address the questions posed by the reviewer. Specific reviewer comments are shown in blue, with the response provided below. Modifications to the original manuscript are highlighted. Comments and responses are grouped thematically.

Reviewer #3 (Remarks to the Author):

In their revision, the authors produced additional analyses and corresponding text to clarify the purpose of the study and to add important additional information. With this increased clarity, some questions emerge.

First, individual gesture classification was done for all available data as described in the manuscript: "Note that this analysis was performed offline, combining data collected from all available open and closed loop control blocks." However, what data was used for the training and testing? Was there cross-validation of the results? The manuscript described the use of LDA for closed-loop decoding but not for the offline gesture classification, although Fig. 7 seems to suggest that there was a nearest neighbor classifier and 10-fold cross validation.

The section titled “**Data-driven bottom-up mapping of the neural latent space of gestures**” describes the how the data from multiple sessions was combined and used for classification offline:

Line 241>> Latent spaces for participant T11 were generated independently for each recording session (with between 1232 and 868 gestures each) and then aligned using a shape-preserving landmark-based procedure ⁴³, resulting in a single matrix representing 7,322 individual gestures as 15D vectors. Subsequent analysis was performed using the entirety of the data for participant T11 aligned in this way.

Line 280>> In order to further characterize the trends observed in the gesture dendrograms, we examined individual trials within the 15D gesture space (Fig. 5). We evaluated the separation between gesture groups using a nearest neighbor classifier (implemented with 10-fold cross validation).

The following section, titled “**Individual gesture Classification**” uses data in a similar way. We have added an additional reference to the previous section to clarify this point:

Line 302>> Note that this analysis was performed offline, combining data collected from all available open and closed loop control blocks. **As described in the previous section, classification was performed using a nearest neighbor classifier (implemented with 10-fold cross validation).**

Second, the manuscript specifies that “for T11, cue onset (the appearance of the instructed gesture, prior to the ‘go’ cue at time zero) was either 1 or 2 seconds before the go cue.” How do the authors explain that, in T11, neural signals begin to change about 2.5 seconds before the go cue (Fig. 3B) and that there is no indication in the results that the data was collected for either 1 or 2 seconds before the go cue (e.g., two peaks in the results in Fig. 3B-D)?

Line 151>> for T11, cue onset (the appearance of the instructed gesture, prior to the ‘go’ cue at time zero) was either 1 or 2 seconds before the go cue, for T5 the delay was 3 seconds.

Figure 3B shows the common dPCA component, which does not carry either gesture or effector (hand) related information. Changes in the common component typically reflect the overall timing of task events, changes in these signals could be interpreted as anticipation of the upcoming gesture cue. Additionally, given the shorter trial time for T11, the 5 second window shown reflects data from more than one trial (we chose to show more data for T11 to match the time window shown for T5, rather than adjusting the time axis to only show one trial). Figure 3C and 3D respectively show the gesture and hand components: in these figures signals begin to change about 1.5 seconds before the go cue for T11, reflecting cue-related information shortly after it becomes available. Figures 3G and 3H show that cue-related signals for T5 begin to

change a bit earlier about 2 seconds before the go cue, as expected for the longer instructed delay period. We have added the following text to clarify these points:

Line 194>> Note that changes in the common component typically reflect the overall timing of task events, including anticipation of upcoming cues. Changes in these components can therefore precede the appearance of instructional cues, but do not reflect either effector or gesture information (as evidenced by the overlap across categories observed in Fig. 3B and F). Gesture and hand components, by contrast, separate by category after cues are provided, reflecting the delay periods employed for each participant (Fig. 3C, D, G and H).

Third, in that vein, Fig. 1C shows that, in T11, neural signals begin to change 1.5 seconds before the go cue, which is clearly different than the time courses shown in Fig. 3. This reviewer understands that the analyses are different, but the underlying neural signals cannot have different time courses unless results for different datasets are shown here or there is some mistake.

Figure 1C was meant to highlight the timecourse of gesture and effector related information after the instruction was provided. The data presented in figure 1C represents the number of features representing activity different from baseline. For this calculation, the baseline values are calculated during the inter trial interval (before the cue is shown). The Kruskal-Wallis statistical test employed finds features with median values that are significantly different from baseline. It is therefore expected that the number of significant features during the inter-trial period should be near zero. dPCA analysis (figure 3) examines the inherent variance represented by different components associated with different marginalization of the data. In this case variance is examined for the entire time period of the task, without an explicit comparison with a designated baseline period. The information derived from the cue is only detected about 200ms after the cue was provided (3C,D,G,H), matching the timecourse observed for Figure 1C. The common component (which does not include cue information) displays a different time course (Fig 3B,F), probably reflecting ramping activity commonly associated with the expectation of upcoming cues.

Fourth, the subjects' task was to begin attempting movements when the falling box crossed the horizontal line, but there is no indication that the subjects followed this instruction as discriminable neural signals clearly begin much earlier than the go cue and classification results are best for data including the time before the go cue. How can the authors exclude the possibility that some of the results described here can be explained by visual information? Additional interpretation/description is needed.

We implemented additional feedback to discourage the participants from attempting the instructed movement before the go cue, as described in the methods:

Line 587>> For CL blocks feedback was provided to the participants based on neural activity in real time: when the cued gesture was decoded, the target and attempt line turned green: if any other gesture (except 'do nothing') was decoded, both objects turned red instead. In order to discourage participants from 'jumping the gun' (attempting the cued gesture before the go cue) the attempt line would turn yellow if the correct gesture was decoded before the target intersected with the attempt line. This last task feature was added for the last three sessions with T11, and both sessions with T5.

We chose the analysis window to encompass the peak in gesture-related activity (Fig. 1C). This window includes neural activity occurring during movement planning as well as the attempted movement. The reviewer correctly points out that the initial part of this response is related to the visual stimulus. This type of response is typically observed during instructed delays when information about the upcoming movement is provided, and has been associated with movement planning. Information related to the instruction becomes evident about 200ms after the visual cue is provided (Fig. 1C, 3C,D,G,H). We have added additional text to clarify this point:

Line 231>> Our analysis focused on activity patterns associated with individual gestures taken from a 2 second window centered on the go cue (Fig.1). **Note that this time window includes activity related to movement planning as well as attempted movement execution. Activity related to movement planning has been widely recognized in motor and premotor cortex⁴³, and has been interpreted as adjusting the state of a dynamical system whose evolution ultimately results in cortical outputs driving movement⁴⁴.**

Fifth, in general, were all analyses reported here done on all data? Fig. 2 talks about classification accuracy in 60 trials (one block) (Fig. 2B-C), but are data in Fig. 2A for all X sets of 6/7 gestures? Please make it clear in the text and the figure legends what data were used to produce the results.

Figure 2 shows data collected during closed loop gesture decoding. We have updated the figure legend to make it clear that each point in 2A corresponds to one block of data using a specific set of 6 or 7 gestures, and 2B shows representative confusion matrices for two of these blocks (the full set of confusion matrices is shown in Supplementary Figure 2).

Line 174>>> Figure 2: Closed-loop decoding accuracy. A. Average decoding accuracy for sets of six (blue) or seven (orange) gestures. Each dot represents the accuracy for a gesture set (i.e. a single block of trials) in a given session. Dotted lines indicate expected chance levels. B,C. Confusion matrices for two sample blocks (with rows representing true classes and columns representing decoded classes), representative of average decoding accuracy for one block in participant T11 (B) and T5 (C). Note that the confusion matrices count trials where no gesture exceeded the likelihood threshold as errors ('no decode', rightmost column). Additional confusion matrices for other gesture sets are included in Supplemental figures S2 and S3.